

# Origin and processing of terrestrial organic carbon in the Amazon system: lignin phenols in river, shelf and fan sediments

Shuwen Sun[1,2,7], Enno Schefuß[2], Stefan Mulitza[2], Cristiano M. Chiessi[3], André O. Sawakuchi[4], Matthias Zabel[2], Paul A. Baker[5,6], Jens Hefter[7], Gesine Mollenhauer[1,2,7]

[1]Department of Geosciences, University of Bremen, Bremen, 28359, Germany
[2]MARUM-Center for Marine Environmental Sciences, University of Bremen, Bremen, 28359, Germany
[3]School of Arts, Sciences and Humanities, University of São Paulo, 03828-000 São Paulo, Brazil
[4]Institute of Geosciences, Department of Sedimentary and Environmental Geology, University of São Paulo, 05508-080 São Paulo, Brazil
[5]Duke University, Nicolas School of the Environment, 301 Old Chemistry, Box 90227, Durham, NC 27708, USA
[6]School of Geological Sciences and Engineering, Yachay Tech, Yachay City of Knowledge, 100650 Urcuqui, Ecuador
[7]Alfred-Wegener-Institute Helmholtz-Zentrum für Polar und Meeresforschung, Bremerhaven, 25570, Germany

*Correspondence to*: Shuwen Sun (shuwen@uni-bremen.de)

**Abstract.** The Amazon River transports large amounts of terrestrial organic carbon ($OC_{terr}$) from the Andean and Amazon neotropical forests to the Atlantic Ocean. In order to compare the biogeochemical characteristics of $OC_{terr}$ in the fluvial sediments from the Amazon drainage basin and in the adjacent marine sediments, we analysed riverbed sediments from the Amazon mainstream and its main tributaries as well as marine surface sediments from the Amazon shelf and fan for total organic carbon (TOC) content, organic carbon isotopic composition ($\delta^{13}C_{TOC}$) and lignin phenol compositions. TOC and lignin content exhibit positive correlations with Al/Si ratios (indicative of the sediment grain size) implying that the grain size of sediment discharged by the Amazon River plays an important role in the preservation of TOC and leads to preferential preservation of lignin phenols in fine particles. Depleted $\delta^{13}C_{TOC}$ values (-26.1 ‰ to -29.9 ‰) in the main tributaries consistently correspond with the dominance of C3 vegetation. Ratios of syringyl to vanillyl (S/V) and cinnamyl to vanillyl (C/V) lignin phenols suggest that non-woody angiosperm tissues are the dominant source of lignin in the Amazon basin. Although the Amazon basin hosts a rich diversity of vascular plant types, distinct regional lignin compositions are not observed. In marine sediments, the distribution of $\delta^{13}C_{TOC}$ and $\Lambda 8$ (sum of eight lignin phenols in organic carbon (OC), expressed as mg/100mg OC) values implies that $OC_{terr}$ discharged by the Amazon River is transported north-westward by the North Brazil Current and mostly deposited on the inner shelf. The lignin compositions in offshore sediments under the influence of the Amazon plume are consistent with the riverbed samples suggesting that processing of $OC_{terr}$ during offshore transport does not change the encoded source information. Therefore, the lignin compositions preserved in these offshore sediments can reliably reflect the vegetation in the Amazon River catchment. In sediments from the Amazon Fan, low lignin content, relatively depleted $\delta^{13}C_{TOC}$ values and high $(Ad/Al)_V$ ratios indicating highly degraded lignin imply that a significant fraction of the deposited $OC_{terr}$ is derived from petrogenic (sourced from ancient rocks) sources.





# 1 Introduction

Rivers deliver annually about 200 Tg of particulate organic carbon (POC) to the oceans (Galy et al., 2007; Ludwig et al., 1996; Schlünz and Schneider, 2000; Galy et al., 2015), which is predominantly deposited on continental shelves and slopes (Hedges and Keil, 1995). Terrestrial organic carbon ($OC_{terr}$) buried in marine sediments has been intensely studied in order to

reconstruct climate and environmental conditions on land (Bendle et al., 2010; Collins et al., 2014; Vogts et al., 2012) with rivers playing a role not only as conduits between terrestrial and marine reservoirs but also as efficient reactors of the $OC_{terr}$ (Aufdenkampe et al., 2011; Battin et al., 2009). During transport in fluvial systems, $OC_{terr}$ is subject to various natural processes, such as selective preservation within the watershed and microbial degradation, as well as anthropogenic processes associated to land-use change (Jung et al., 2014; Wang et al., 2015; Wu et al., 2007). In marine environments, $OC_{terr}$ mixes

with marine organic carbon and experiences further diagenetic alteration (Aller and Blair, 2006). The refractory fractions of $OC_{terr}$, which survive these processes, are preserved at sites of sediment deposition. As a result of this extensive processing, the climatic and environmental information recorded by $OC_{terr}$ in marine sediments may be subject to temporal and spatial offsets. Hence, a comparison between the characteristics of $OC_{terr}$ in drainage basins and adjacent continental margins is helpful to decipher which depositional sites can provide reliable marine sedimentary archives and to what extent they reflect

climatic and environmental changes within the catchment.

The Amazon River is of special interest due to its large drainage basin size and discharge of $OC_{terr}$ to the ocean. Previous studies that assessed the fate of $OC_{terr}$ transported and discharged by the Amazon mainly relied on bulk organic parameters, isotope compositions of total organic carbon ($\delta^{13}C_{TOC}$ and $\Delta^{14}C_{TOC}$) (Bouchez et al., 2014) as well as on analyses of various biomarkers (e.g., lignin, plant-waxes and tetraether lipids) (Zell et al., 2014). Based on the correlation between TOC contents

and Al/Si ratios, the latter being indicative of grain-size variations, Bouchez et al. (2014) showed a distinct mineral and size class association of particulate organic matter (POM), which in turn affects its transport in the fluvial system. Moreover, the $\delta^{13}C_{TOC}$ and $\Delta^{14}C_{TOC}$ of POM indicated that rock-derived POM accounts for a significant proportion of riverbed sediments. Hedges et al. (1986) reported lignin compositions of typical plant tissues in the Amazon basin and found that POM has distinct lignin compositions in different grain size fractions.

Offshore the Amazon River mouth, the predominant depo-center of terrestrial organic matter has been shown to change between glacial and interglacial periods (Schlünz et al., 1999). During glacials (i.e., low sea-level), most of the Amazon derived $OC_{terr}$ is deposited on the Amazon Fan, while during interglacials (i.e., high sea-level), along-shore currents result in deposition of $OC_{terr}$ on the continental shelf northwest of the Amazon River mouth. Terrestrial climate and vegetation from the last glacial period has thus been reconstructed by analysing molecular biomarkers, such as lignin, plant-wax lipids, and

branched glycerol dialkyl glycerol tetraethers (GDGTs) in sediment cores recovered from the Amazon Fan (Bendle et al., 2010; Boot et al., 2006; Goñi et al, 1997; Kastner and Goñi, 2003)

Lignin is a useful tracer for $OC_{terr}$ because it is exclusively produced by vascular plants and accounts for about 20-30 % of dry biomass in woody plants (Zhu and Pan, 2010) and 15-20 % in grasses (Perez-Pimienta et al., 2013). It is relatively





resistant to microbial degradation (Killops and Killops, 2005) and abundant in many environments (e.g. Kuzyk et al., 2008; Loh et al., 2012; Winterfeld et al., 2015). To date, little is known about the factors influencing lignin composition in the Amazon basin and adjacent marine sediments, its transport pathways in the Amazon continental margin, and the potential of lignin in offshore sedimentary archives to constrain sources and compositions of $OC_{terr}$.

Here we determined lignin contents and composition in riverbed sediments of the Amazon River and its lowland tributaries as well as in surface sediments of the Amazon shelf and slope. By doing so, we provide evidence on the spatial distribution of $OC_{terr}$, its plant source, its origin within the catchment, and its dispersal patterns on the Amazon continental margin.

## 2 Study area

The Amazon River originates from the confluence of the Ucayali and Marañón Rivers in the Andean region in southwestern
Peru and receives numerous tributaries that form the largest hydrographic basin in the world (Goulding et al., 2003). It covers an area of about $6.1 \times 10^6$ km$^2$ extending from the Guiana Highlands in the north to the Central Brazil Highlands in the south, and is bordered by the Andes mountain range in the west (Guyot et al., 2007). The Peruvian and Bolivian Andean tributaries Solimões (upper stretch of the Amazon River from its confluence with the Negro River) and Madeira are typical white water rivers. Because they drain the steep slope and rapidly weathering Andean region, they are characterized by high
concentrations of suspended sediments and dissolved nutrients (Gibbs, 1967). The other major tributaries in the Amazon basin drain lowland regions and are classified as either black or clear water rivers. The black water rivers, such as the Negro River, are rich in dissolved humic substances derived from podzols and depleted in suspended sediments (Mounier et al., 1999). The clear water rivers (e.g., the Xingu River) have low concentrations of suspended sediments and dissolved organic matter compared to the white and black water rivers, and their clarity allows high phytoplankton productivity (Junk, 1997;
Richey et al., 1990). Although the deforested area of the Amazon basin is increasing significantly in the eastern and south-eastern portions, the remainder of the lowland Amazon basin is largely forested except for some small areas dominated by savannah (Houghton et al., 2001). Elsewhere, some grasses grow along the shoreline regions of white water rivers (Guyot et al., 2007; Hedges et al., 1986).

The Amazon basin annually receives an average of about 2500 mm rainfall and has the world's largest water discharge of
about $2 \times 10^5$ m$^3$s$^{-1}$ (Callede et al., 2000; Guyot et al., 2007). Up to 40 Tg of carbon are discharged along with $8\text{-}12 \times 10^{11}$ kg suspended sediment load each year by the Amazon River into the Atlantic Ocean, which makes the Amazon the largest fluvial source of $OC_{terr}$ to the ocean (Dunne et al., 1998; Moreira-Turcq et al., 2003). The Amazon-derived plume of water and suspended sediment is advected northwestward along the northern South American coastline by the North Brazil Current, eventually forming the Amazon subaqueous delta-Guianas mud belt extending 1600 km along the northeastern
coast of South America (Geyer et al., 1996; Nittrouer and DeMaster, 1996). The Amazon Fan is located off the northern coast of Brazil centered around 4 °N, extending 700 km from the shelf break seaward reaching a maximum width of about 650 km. The Amazon Fan is largely inactive today, but during past periods of low sea level such as the Last Glacial



Maximum, large amount of suspended sediment and bed load were transported via submarine canyons and deposited on the Fan (Schlünz et al., 1999).

## 3 Materials and methods

### 3.1 Sample collection

Riverbed sediments were collected from the Amazon River mainstream and its main tributaries during two sampling campaigns in November 2011 and May 2012, corresponding respectively to the dry and wet seasons. Sediment samples were retrieved from sites with different channel depths to better represent their range of grain size variability. A Van Veen grab sampler was used for sampling, and the station locations are shown in Table 1 and Fig. 1A.

Marine surface sediments from the Amazon shelf and fan were collected during two cruises (Table 2, Fig. 1B). The GeoB

samples were recovered with a multicorer in February/March 2012 during the R/V Maria S. Merian cruise MSM 20/3, while the other marine surface sediments were taken in February/March 2010 with a box corer deployed from R/V Knorr during cruise KNR197-4. All samples were kept frozen at -20 °C before analysis and were subsampled into 1 cm intervals. The uppermost 2 cm of GeoB multicore samples and slices of 1 cm from intervals between 5 and 8 cm sediment depth of cores taken during cruise KNR197-4 were used in this study.

The GeoB surface sediments were oven dried at 50 °C, while the riverbed and the other marine surface sediments were freeze dried in a Christ Alpha 1-4 LD plus freeze dryer. After drying, all samples were homogenized for further analysis.

### 3.2 Grain size analysis

Grain size analysis was only conducted for marine sediments, because the riverbed sediment samples had already been ground before sub-sampling. For grain size measurement of the terrigenous fraction, bulk marine sediments were pre-treated

as follows. Samples of about 0.5 g were successively boiled with $H_2O_2$ (35 %), HCl (10 %) and NaOH to remove respectively organic matter, carbonate and biogenic silica. To prevent potential aggregation, 10 ml of dissolved sodium pyrophosphate ($Na_4P_2O_7 \cdot 10H_2O$) was added immediately prior to grain size analysis. Samples were measured using a Laser Diffraction Particle Size Analyser (Beckman Coulter laser particle sizer LS-13320) in 116 size classes ranging from 0.04 to 2000 μm. All measurements were performed in demineralized and degassed water to avoid interference of gas bubbles.

### 3.3 Elemental and bulk isotopic analysis

About 4 g dry sediment was used to measure the Al and Si elemental concentrations by energy dispersive polarization X-ray fluorescence (EDP-XRF) spectroscopy. The device was operated with the software Spectro X-Lab Pro (Version 2.4) using the Turboquant method. Analytical quality of measurements was assessed by repeated analyses of the certified standard reference material MAG-1, and the standard deviation of replicate measurement of sediment samples was better than 0.5%.





After removal of carbonates with 12.5 % HCl and drying, the total organic carbon content (TOC) of all samples was determined by a LECO CS 200 CS-Analyzing System. The relative standard deviation of duplicate analyses was better than 1%.

Stable carbon isotopic ratios of TOC ($\delta^{13}C_{TOC}$) were analyzed on a Finnigan MAT Delta plus coupled with a CE elemental analyzer and a Con-Flo II interface. Samples were pre-treated by the same method as used for the TOC measurement. $\delta^{13}C_{TOC}$ values were reported using the standard notation relative to the Vienna Pee Dee Belemnite (VPDB) standard. The uncertainty was less than ±0.1‰, as calculated by long-term repeated analyses of the internal reference sediment (WST2).

### 3.4 Lignin-phenol analysis

Alkaline CuO oxidation was used to obtain eight lignin-derived phenols (vanillyl phenols, syringyl phenols and cinnamyl phenols) and three para-hydroxybenzenes. A CEM MARS5 microwave accelerated reaction system was used to perform alkaline CuO oxidation of lignin based on the approach described by Goñi and Montgomery (2000). Dried sediment samples (containing about 2-5 mg of TOC) were oxidized with CuO (500 mg) and ferrous ammonium sulfate (50 mg) in de-aerated 2 N NaOH at 150 °C for 90 min under a nitrogen atmosphere. After the oxidation, known amounts of recovery standards (ethyl vanillin and *trans*-cinnamic acid) were added to each reaction tube. The alkaline supernatant was transferred and acidified to pH 1 by addition of concentrated HCl. Reaction products were extracted twice with ethyl acetate and water in the ethyl acetate solution was removed by addition of $Na_2SO_4$. Ethyl acetate was evaporated under a continuous nitrogen flow. Once dry, 400 μl of pyridine was added immediately to re-dissolve the reaction products. Lignin phenols were analyzed by gas chromatography-mass spectrometry (GC-MS). Prior to the injection to the GC-MS, compounds in pyridine were derivatized with bis-trimethylsilyl-trifluoroacetamide (BSTFA) +1 % trimethylchlorosilane (TMCS) to silylate exchangeable hydrogen. Chromatographic separation was achieved by a 30 m×0.25 mm (i.d.) DB-1MS (0.25 μm film thickness) capillary GC column. The temperature program was 100 °C initial temperature, 4 °Cmin$^{-1}$ ramp and 300 °C final temperature with a hold of 10 minutes.

The eight lignin-derived phenols obtained by alkaline CuO oxidation are subdivided into the following groups: the vanillyl (V) phenols include vanillin (Vl), ancetovanillone (Vn), and vanillic acid (Vd); syringyl (S) phenols are syringaldehyde (Sl), acetosyringone (Sn) and syringic acid (Sd); and cinnamyl (C) phenols consist of *p*-coumaric acid (*p*-Cd) and ferulic acid (Fd). The para-hydroxybenzenes include *p*-hydroxybenzaldehyde (Pl), *p*-hydroxybenzophenone (Pn) and *p*-hydroxybenzoic acid (Pd). The C phenols are only present in non-woody tissues of vascular plants while the S phenols are unique to angiosperms and V phenols exist in all vascular plants. Therefore, the ratios of S/V and C/V divide the plant sources of lignin into four types, non-woody and woody tissues of gymnosperms and angiosperms (Hedges and Mann., 1979a). (Ad/Al)$_V$ and (Ad/Al)$_S$ refers to the acid to aldehyde ratios of V and S phenols, which indicate the degradation of lignin (Ertel and Hedges, 1985).

The lignin phenols and identified compounds were quantified by individual response factors calculated from mixtures of commercially available standards analyzed periodically. The yields of Pl, Vl and Sl were calculated by the recovery rate of



ethyl vanillin and the recovery rate of trans-cinnamic acid was applied for the yield estimation of other lignin-derived compounds (Kuzyk et al., 2008).

Carbon-and sediment-normalized lignin yields are respectively reported as Λ8 (mg/100mg OC) and Σ8 (mg/10g dry sediment). These measures indicate respectively the relative contributions of vascular plant material to the TOC, and to the total samples, and were respectively calculated as the sum of S, V and C phenols in 100 mg organic carbon and 10 g dried sample (Hedges and Mann, 1979b).

## 4 Results

### 4.1 Riverbed sediments

#### 4.1.1 TOC and stable carbon isotopic composition

Riverbed sediments contain between 0.13 % and 3.99 % dry weight (wt) TOC (Table 1, Fig. 2A). Negro River and Xingu River sediments have similar TOC contents (0.53 %-3.99 %, mean=2.08±1.15 %, $n$=7 and 0.52 %-3.82 %, mean=2.44±1.37 %, $n$=8, respectively), which are higher than the TOC contents in other main tributaries. Consistently low TOC contents were observed in the Madeira River varying from 0.14 % to 0.52 % (mean=0.36±0.17 %, $n$=6). Solimões River sediments display TOC contents ranging from 0.28 % to 0.90 % with an average of 0.62±0.20 % ($n$=7). Amazon River mainstream sediments have intermediate TOC contents ranging from 0.13 % to 1.44 % (mean=0.73±0.36 %, $n$=19).

The range of $\delta^{13}C_{TOC}$ values of all riverbed samples is from -26.1 ‰ to -29.9 ‰ (Table 1, Fig. 2B). The most enriched (-26.1 ‰) and depleted (-29.9 ‰) $\delta^{13}C_{TOC}$ values are both observed in Solimões River sediments, which have an average value of -28.2±1.2 ‰ ($n$=6). A similar scatter of $\delta^{13}C_{TOC}$ values is obtained from sediments of the Negro River varying between -26.5 ‰ and -29.8 ‰ (mean=-28.7±1.3 ‰, $n$=7). In comparison, narrower ranges are found in the other tributaries and the Amazon River mainstream, with values ranging from -27.8 ‰ to -28.4 ‰ (mean=-28.0±0.3 ‰, $n$=3) for the Madeira River, from -27.9 ‰ to -29.8 ‰ (mean=-29.2±0.8 ‰, $n$=8) for the Xingu River, and from -27.5 ‰ to -29.4 ‰ (mean=-28.2±0.5 ‰, $n$=15) for the Amazon River mainstream.

### 4.1.2 Lignin phenols

The Λ8 values in riverbed sediments vary from 0.73 mg/100mg OC in the Madeira River to 9.27 mg/100mg OC in the northern channel of the Amazon River mouth (Table 1, Fig. 2C). The highest average Λ8 of 4.92±1.44 mg/100mg OC ($n$=8) is observed in the Xingu River with values ranging between 3.30 and 6.91 mg/100mg OC. In contrast, Madeira River sediments display the lowest average Λ8 value of 2.61±1.30 mg/100mg OC ($n$=6; 0.73 to 4.42 mg/100mg OC). The Λ8 values of samples in the Negro River and the Solimões River vary respectively between 3.55 and 4.89 mg/100mg OC (mean=4.38±0.51 mg/100mg OC, $n$=7) and 3.49 and 6.44 mg/100mg OC (mean=4.59±0.96 mg/100mg OC, $n$=7). The Λ8



values of the mainstream sediments have the largest variability, from 0.75 to 9.27 mg/100mg OC (mean=4.88±1.93 mg/100mg OC, $n$=19).

Negro River and Xingu River sediments have higher Σ8 values and greater variability, ranging respectively from 1.89 to 19.52 mg/10g dry sediment (mean=9.50±5.90 mg/10g dry sediment, $n$=7) and from 2.73 to 17.97 mg/10g dry sediment 5 (mean=11.18±5.93 mg/10g dry sediment, $n$=8). In contrast, the samples from the other tributaries and the Amazon mainstream display lower Σ8 values, 0.99-4.94 mg/10g dry sediment (mean=2.91±1.22 mg/10g dry sediment, $n$=7) for the Solimões River, 0.10-2.30 mg/10g dry sediment (mean=1.11±0.85 mg/10g dry sediment, $n$=6) for the Madeira River and 0.10-11.05 mg/10g dry sediment (mean=3.96±2.93 mg/10g dry sediment, $n$=19) for the Amazon River mainstream. Apart from the Xingu River, the Σ8 values and Λ8 values in individual tributaries are well correlated ($r^2$ from 0.64 to 0.96, 10 $p$<0.05).

S/V and C/V of all riverbed samples range respectively from 0.70 to 1.51 and 0.08 to 0.47 (Table 1, Fig. 3). Negro River and Xingu River sediments show slightly lower average S/V ratios (0.88 and 1.00) than samples from the Solimões River, the Madeira River and the Amazon mainstream (1.08, 1.10, and 1.09, respectively). The range of average C/V ratios of all tributaries is from 0.16 to 0.24 and the highest mean C/V ratio was observed in the Madeira River. Except for the highest 15 P/V ratio in the Madeira River (0.38), the P/V ratios in the other tributaries varied only slightly (0.24-0.27). $(Ad/Al)_V$ values in all riverbed samples (Table 1, Fig. 2D) vary from 0.26 to 0.71 with an average of 0.42, and $(Ad/Al)_S$ values range from 0.15 to 0.57 with an average of 0.28. The $(Ad/Al)_V$ and $(Ad/Al)_S$ are correlated ($r2$=0.76, $p$<0.05, $n$=47), but all $(Ad/Al)_S$ values were lower than the respective $(Ad/Al)_V$ values.

### 4.1.3 Al/Si

20 In riverbed sediments, the Al/Si varies from 0.11 to 0.56 (Fig. 4A), which is larger than the Al/Si range of riverbed sediments reported by Bouchez et al. (2011). Solimões River sediments have a narrow Al/Si range (0.27-0.37) with an average of 0.33±0.04. The samples in the Madeira River and the Amazon River mainstream display lower Al/Si values (0.17-0.37, mean=0.26±0.08, $n$=6 and 0.11-0.38, mean=0.29±0.07, $n$=19, respectively). On the contrary, Negro River and Xingu River sediments have larger variations and higher average Al/Si ratios (0.17-0.51, mean=0.37±0.15, $n$=7 and 0.14- 25 0.56, mean=0.40±0.15, $n$=8, respectively).

### 4.2 Marine sediments

### 4.2.1 TOC and stable carbon isotopic composition

TOC content in the marine surface sediments on the Amazon shelf varies from 0.11 % to 0.98 % (Table 2, Fig. 5A). Three samples near the Amazon River mouth exhibit low TOC contents of less than 0.2 % (0.13±0.04 %, GeoB16209-2, 30 GeoB16210-2 and GeoB16211-2). The highest TOC content is obtained for BC61C, the southernmost sample from the Amazon Fan. The $\delta^{13}C_{TOC}$ values for marine surface sediments range from -18.6 ‰ at station GeoB16202-1, which is





located southeast of the river mouth (SE sector), to -26.7 ‰ at station MC33A (near the river mouth) (Table 1 and Fig. 5B). The samples from the SE sector reveal the most enriched $\delta^{13}C_{TOC}$ values between -18.6 ‰ and -21.6 ‰ (-19.6±1.1 ‰, $n$=6). The samples collected off the Amazon River mouth exhibit the most depleted $\delta^{13}C_{TOC}$ values (-24.8±1.4‰, $n$=5). $\delta^{13}C_{TOC}$ values increase along the shelf towards the northwest from the Amazon River mouth to around -20.4 ‰ at the north-

5 westernmost stations deeper than 2000m. Apart from the relatively depleted $\delta^{13}C_{TOC}$ value of -24.5 ‰ at station BC61C, the $\delta^{13}C_{TOC}$ values on the Amazon Fan range between -21.4 ‰ and -23.0 ‰.

### 4.2.2 Lignin phenols

The Λ8 values in marine surface sediments vary from 0.04 to 2.01 mg/100 mg OC (Table 2, Fig. 5C). The samples in the SE sector display very low Λ8 values (0.04-0.17 mg/100mg OC), which increase slightly with distance from the Amazon River

mouth. Apart from the higher lignin content at station BC61C (0.57 mg/100mg OC), the samples in the Amazon Fan (Fan sector) also have low Λ8 values ranging from 0.05 to 0.22 mg/100mg OC with a decreasing trend with distance from the Amazon River mouth. The distribution pattern of lignin content in the northwest area (NW sector) is similar to that shown by $\delta^{13}C$ values. In the NW sector, Λ8 values increase first from 0.19 to 2.01 mg/100mg OC at stations closest to the Amazon River mouth, decrease northwestward and reach rather constant low values of 0.18 mg/100mg OC on the offshore slope of

the NW sector. The Σ8 values in marine surface sediments vary from 0.01 to 1.49 mg/10g dry sediment and are highly correlated with Λ8 values ($r^2$=0.85, $p$<0.05, $n$=30).

Ranges of S/V and C/V ratios in marine sediments are similar to those observed in riverbed sediments. S/V and C/V ratios vary respectively from 0.59 to 1.62 and from 0.10 to 0.43 (Table 2, Fig. 3). Sediments in the SE sector have higher C/V ratios (mean=0.30±0.11, $n$=9). The $(Ad/Al)_V$ and $(Ad/Al)_S$ values in marine samples vary respectively from 0.37 to 1.16

(Table 2, Fig. 5D) and from 0.32 to 0.97 (Table 2). However, our marine samples show no correlation between $(Ad/Al)_V$ and $(Ad/Al)_S$ values. Samples from the Amazon Fan exhibit the highest average $(Ad/Al)_{V,S}$ values of 0.81 and 0.71. Samples in the SE and NW sectors have similar average $(Ad/Al)_V$ values (0.62 and 0.61, respectively), but samples from the NW region exhibit larger variation.

### 4.2.3 Al/Si and grain size

In marine sediments, the Al/Si varies from 0.14 to 0.47. The SE and Fan sectors have narrow Al/Si ranges, which are respectively 0.37-0.41 with an average of 0.39±0.01 and 0.40-0.46 with an average of 0.43±0.02 (Table 2.). In contrast, the samples in the NW sector display large Al/Si variance (0.14-0.47, mean=0.32±0.14).

Grain size of marine sediments is reported as mean grain size and varies between 3.23-92.90 μm (Table 2). Coincident with the Al/Si ratio distribution, samples in the SE and Fan sectors respectively have narrow grain size ranges, which are 4.44-

30 11.79 μm with an average of 8.07±2.75 μm and 3.23-14.50 μm with an average of 5.84±3.38 μm. On the contrary, sediments in the NW sector display wide range grain sizes (4.74-92.90 μm, mean=31.23±35.19 μm). The correlation between grain size



and Al/Si ratio ($r^2$=0.85, $p$<0.05, $n$=28) indicates that the Al/Si ratio is a reliable proxy for the grain size of terrestrial sediments.

## 5 Discussion

### 5.1 Lowland Amazon river system

#### 5.1.1 Spatial distribution and isotopic composition of $OC_{terr}$

The TOC contents of our riverbed sediments in individual tributaries are larger than values of bedload sediments reported by Bouchez et al. (2014) (mean=0.23±0.42 % in the Solimões and Madeira rivers and the Amazon mainstream), but lower than the values of suspended materials (e.g. mean=1.14±0.33 % in the Solimões and Madeira rivers and the Amazon mainstream) in the Amazon lowland basin (Hedges et al., 1986; Moreira-Turcq et al., 2003; Bouchez et al. , 2014). The distribution of

TOC contents basically reflects the characteristics of the tributaries (Fig. 2A). The relatively high TOC contents in the Negro River are due to the low suspended sediment content and high content of humic substances (Ertel et al., 1986). The Xingu river is characterized by low suspended sediment content and high phytoplankton production, which lead to the high TOC contents in riverbed sediments (Moreira-Turcq et al., 2003). In contrast, the Solimões River and the Madeira River being the primary contributors of the suspended sediment to the Amazon River mainstream, have large suspended sediment load

(Moreira-Turcq et al., 2003). Consequently, the low TOC contents in the riverbed sediments in these tributaries are due to dilution by lithogenic material. The lower-intermediate TOC contents for the Amazon River mainstream results from the mixing of different signals from these tributaries with a greater influence from the Solimões and the Madeira Rivers.

The $\delta^{13}C_{TOC}$ values of our riverbed sediments (i.e., from -26.1 ‰ to -29.9 ‰) (Fig. 2B) are similar to the values reported for riverbed sediments (e.g. from -27.6 ‰ to -28.8 ‰ ) and suspended particulate matter (e.g. -28.3±1.1 ‰) in the Amazon

River system in previous studies (Hedges et al., 1986; Cai et al., 1988; Kim et al., 2012; Bouchez et al. ,2014). Hedges et al. (1986) studied $\delta^{13}C_{TOC}$ values of different organic carbon sources in samples from the Amazon River and found the respective average $\delta^{13}C_{TOC}$ values of C3 tree leaves, woods, macrophyte tissues and C4 grasses to be -30.1±0.9 ‰, -27.6±1.0 ‰, -21.4±8.4 ‰ and -12.2 ‰. The total average $\delta^{13}C_{TOC}$ value of riverbed sediments in this study (-28.5±0.9 ‰) confirm the dominant contribution from terrestrial C3 plants. There is no significant difference in the distribution of $\delta^{13}C_{TOC}$ values

among the sampled tributaries.

#### 5.1.2 Characteristics of lignin phenols

With the exception of the samples from the Xingu River, all riverbed sediments exhibit a good relation between Λ8 values and Σ8 values (average $r^2$=0.76, $p$<0.05, $n$=39). In the Xingu River, in contrast, this relation was weak likely due to the higher phytoplankton production (Moreira-Turcq et al., 2003), which changes Λ8 but not Σ8 (Rezende et al., 2010). Based

on previous studies, the Λ8 values of different organic matter fractions range from 0.45 to 2.40 mg/100 mg OC for fine



particulate organic matter (FPOM, silt and clay fraction, <63 μm), and from 1.21 to 10.46 mg/100 mg OC for coarse particulate organic matter (CPOM, sand-size fraction, >63 μm) (Aufdenkampe et al., 2007; Hedges et al., 1986; Hedges et al., 2000). In this study, riverbed sediments (Fig. 2C), except for three samples with lignin contents lower than 2.0 mg/100 mg OC, had $\Lambda 8$ values (2.42-9.27 mg/100 mg OC) similar to those of CPOM. Most of the $\Lambda 8$ values of our samples are

smaller than the average $\Lambda 8$ values of tree wood tissues (19.3 mg/100 mg OC) and C4 grasses (9.1 mg/100 mg OC), and closer to the range of tree leaves and macrophytes (3.7 mg/100mg OC and 6.4 mg/100mg OC, respectively) (Hedges et al., 1986). This finding is also supported by the distribution of C/V and S/V ratios. The plot of S/V vs. C/V (Fig. 3) indicates that angiosperm leaves are the major origin of lignin in the lower Amazon basin. It is noteworthy that the range of typical C/V values for angiosperm leaf material in the Amazon basin is larger (i.e., including C/V values as low as 0.07) than in other

regions (with lowest C/V values around 0.20) (Bianchi et al., 2011; Cathalot et al., 2013; Tesi et al., 2014). The resulting small difference between C/V ratios of non-woody and woody tissues of angiosperms in the Amazon region results in a larger uncertainty in inferring the plant sources of lignin. C/V values around 0.1 could be interpreted either as signals exclusively from leaves or as signals from a mixture of woody tissues and leaves. To circumvent this uncertainty, the P/V values are also used to identify the lignin sources. P phenols in our samples are derived from lignin, as supported from the

significant correlation of the content of P phenols and lignin content ($r^2$=0.50, $p$<0.05, $n$=47). All P/V values of our samples (0.17-0.64) are higher than the average P/V ratio of woods (0.05) and similar to the range observed for leaves (0.16-6.9) (Hedges et al., 1986). Considering all parameters, non-woody angiosperms are the most likely major source of lignin in the lowland Amazon basin. The slightly higher C/V ratios in the Solimões River (0.20) and the Madeira River (0.24) suggest a small contribution of grass-derived material (C/V>1) probably from the Andean highlands (Aufdenkampe et al., 2007;

Hedges et al., 2000).

The degradation extent of $OC_{terr}$ can be assessed by $(Ad/Al)_V$ and $(Ad/Al)_S$ ratios as more degraded lignin yields elevated $(Ad/Al)_V$ and $(Ad/Al)_S$ values (Hedges et al., 1988; Opsahl and Benner, 1995). In the case of the Amazon basin, the $(Ad/Al)_V$ and $(Ad/Al)_S$ ratios of typical fresh woods and tree leaves both range from 0.11 to 0.24 (Hedges et al., 1986). All of our samples exhibiting values between 0.26 and 0.71 for $(Ad/Al)_V$ (Fig. 2D) and between 0.15 and 0.57 for $(Ad/Al)_S$ are outside

of the range of fresh plant materials, suggesting degraded $OC_{terr}$ in all samples. Instead, the $(Ad/Al)_{V,S}$ ratios observed in our samples are within the ranges of suspended particulate solids obtained in the lower Amazon basin and Bolivian headwaters ($(Ad/Al)_{V,S}$ of 0.21-0.39 and 0.13-0.22 for CPOM and $(Ad/Al)_{V,S}$ of 0.38-0.79 and 0.22-0.41 for FPOM (Hedges et al., 1986; Hedges et al., 2000)). The Negro River displayed the highest average $(Ad/Al)_V$ ratio (0.55), reflecting a greater degree of degradation. This might be indicative of more efficient degradation in the podzols of the lateritic landscapes in the Negro

River watershed (Bardy et al., 2011). The $(Ad/Al)_V$ ratios in the Solimões and the Madeira Rivers increase with increasing C/V values ($r^2$=0.50, $p$<0.05, $n$=13), which implies that the plant tissues with higher C/V values (higher grass contributions) are more degraded. This further supports the inference that the Solimões and the Madeira Rivers receive POC from grass sources from high-altitude watersheds, where deeper soil erosion of more degraded $OC_{terr}$ could occur. The degradation status of lignin in riverbed sediments does not display a downstream increasing trend and is similar to previous studies on





suspended POC of different size fractions. This leads to the conclusion that $OC_{terr}$ processing during transport through the Amazon river system is minimal and the degradation information likely reflects source characteristics of $OC_{terr}$ prior to fluvial transport (Hedges et al., 1986; Hedges et al., 1994).

### 5.1.3 Sedimentological control on $OC_{terr}$ characteristics

As grain size data could not be obtained on the riverbed sediment samples, we inferred grain size information based on the relationship between the Al/Si ratio and grain size of riverbed sediments observed by Bouchez et al. (2011) in samples from the Amazon basin. High Al/Si indicates aluminium-rich fine-grained sediment, whereas low Al/Si suggests silicon-rich particles of larger grain size (Bouchez et al., 2011; Galy et al., 2008).

As expected, the TOC contents increase with Al/Si (Fig. 4A), indicating that fine particles, associated with larger specific

surface areas and likely rich in clay, carry more TOC than coarser particles. The Negro and the Xingu Rivers have larger Al/Si and TOC variations, and for a given Al/Si ratio, the Negro and the Xingu Rivers show higher TOC contents compared to the other tributaries. As these rivers have distinct chemical characteristics and clay mineral composition (e.g., lower pH in the waters of the Negro River and higher kaolinite content in the sediments of the Negro and the Xingu Rivers than in other tributaries; Guyot et al., 2007), the adsorption affinities of $OC_{terr}$ on different clay minerals or under different chemical

conditions may be distinct.

Different grain size classes may not only have different TOC content but also the composition of their $OC_{terr}$ might vary. For example, previous studies on POM in the Amazon basin found that CPOM has a higher content of lignin phenols than FPOM and that CPOM is composed of fresher lignin with lower C/V ratios (Hedges et al., 1986). Nevertheless, contradictory results were observed in our riverbed sediments. The Λ8 values in the Madeira River, the Solimões River and

the mainstream Amazon River show a remarkable increase with decreasing grain size (indicated by increasing Al/Si ratios) (Fig. 4B). This rise in lignin content in organic matter associated with finer minerals implies preferential preservation of lignin on finer particles compared with other components. In the Negro River, there is only a slight increase in Λ8 values as mineral particles become finer, probably as a result of the large amount of sediment-associated chemically stable humic substances (Hedges et al., 1986), in which the lignin content is relatively constant. However, Xingu River sediments exhibit

decreasing Λ8 values with decreasing grain size probably because the lignin content in finer particles from the Xingu River is diluted by other non-lignin organic components. With respect to the indicator of plant sources (Fig. 4C), the C/V ratios for samples from the Madeira River, the Solimões River and the Negro River decrease with decreasing grain size, which implies that lignin with higher C/V ratios is typically enriched in coarser particles. This suggests that the non-woody tissues with higher proportions of cinnamyl phenols are enriched in coarse-grained sediments. Xingu River and Amazon River

mainstream sediments present no pronounced trend between C/V ratios and Al/Si values.

$(Ad/Al)_V$ values for all riverbed sediments do not show any obvious relationship with Al/Si (Fig. 4D). Only Madeira River and Solimões River sediments exhibit decreasing $(Ad/Al)_V$ values with increasing Al/Si, which suggests that lignin associated to larger mineral particles is more degraded. This observation supports the preferential preservation of lignin in



finer-grained sediments due to better protection against oxidative degradation (Killops and Killops, 2005). In previous studies on suspended sediments, lignin in the coarse fractions is more abundant and less degraded compared to the counterpart in fine fractions (e.g. Hedges et al., 1986). In contrast, our results for riverbed sediments suggest that lignin is preferentially preserved and better protected against degradation on fine-grained material. The different grain size effects on

5  $OC_{terr}$ composition between suspended and riverbed sediments suggest that there are other processes working on $OC_{terr}$ in suspended sediments and riverbed sediments which cause post-depositional changes in the $OC_{terr}$ characteristics.

In summary, our data indicate that lignin derives mainly from non-woody tissues of angiosperms in the lowland Amazon basin, and there is little evidence for contribution from C4 plants to riverbed sediments. Grain size plays an important role in $OC_{terr}$ preservation and lignin composition in the Amazon River. Fine inorganic particles have high adsorption affinity for

10  $OC_{terr}$, especially for lignin compared to other $OC_{terr}$ components and efficiently protect lignin from degradation.

### 5.2 Amazon shelf and fan

### 5.2.1 Spatial distribution and characteristics of $OC_{terr}$ and lignin phenols

Because of the depleted average $\delta^{13}C_{TOC}$ values of the riverbed sediments (-28.5±0.9 ‰), contribution of C4 plants is not expected in the offshore sediments affected by the Amazon outflow. Therefore, enriched $^{13}C_{TOC}$ values in the SE sector (-

15  18.6 ‰ to -21.6 ‰) likely indicate organic matter predominantly of marine origin. $\delta^{13}C_{TOC}$ values in the Amazon Fan sector ranging from -21.4 ‰ to -24.5 ‰, (Fig. 5B) also reflect dominantly marine organic matter. These values are within the range of published values for high sea-level periods (Schlünz et al., 1999), when most of terrestrial POM discharged from the Amazon River is transported to the north-western shelf. In the NW sector, increasing $\delta^{13}C_{TOC}$ values with distance from the Amazon River mouth indicate that the terrestrial POM input from the Amazon River transported to the NW by the North

20  Brazil Current is increasingly diluted by marine organic matter. $OC_{terr}$ is dominant on the continental shelf, corroborating previous results (e.g., Schlünz et al., 1999).

Sediments in the SE sector exhibit much lower Λ8 values than observed in the Fan and NW sectors. The Λ8 values in sediments near the Amazon River mouth are highly variable and decrease with distance from the river mouth to the Fan and NW sectors, reaching very low Λ8 value in the slope of the NW sector. Λ8 and $\delta^{13}C_{TOC}$ values show similar spatial

25  distribution and are positively correlated (Fig. 6). The agreement in the spatial patterns of lignin content and isotope composition of organic matter suggest that lignin is a reliable tracer of $OC_{terr}$ in the Amazon shelf and fan, and that the SE sector receives little $OC_{terr}$ contribution from the Amazon River. The intercept of the correlation between Λ8 and $\delta^{13}C_{TOC}$ of NW and Fan sediments is at -20.8 ‰, which corresponds to conditions with minimal $OC_{terr}$ input to the marine sediments. It should be noted that the samples from the Amazon Fan have the same low lignin contents as in the SE sector, which

30  indicates low contribution of $OC_{terr}$ from terrestrial vascular plants under modern conditions. However, the Fan sediments show more depleted $\delta^{13}C_{TOC}$ than sediments from the SE sector, which implies that a small terrestrial fraction is contained in the organic matter of the modern Fan sediments. Potentially, the $OC_{terr}$ from vascular plants deposited in the Amazon Fan is





readily degraded as indicated, e.g., by the high $(Ad/Al)_{V,S}$ ratios, while the relict $OC_{terr}$ is predominantly rock-derived (with estimated $\delta^{13}C_{TOC}$ between -24.3 ‰ and -25.7 ‰) (Bouchez et al., 2014) and responsible for the depleted $\delta^{13}C_{TOC}$. Petrogenic organic matter is thus likely a significant component in the offshore sediments because of its refractory nature and resulting high preservation potential.

C/V and S/V ratios (0.08-0.47 and 0.70-1.57, respectively; Fig. 3) in the entire Amazon shelf and fan are comparable to those in the riverbed sediments of the lowland Amazon basin, which indicates the same predominant source of non-woody angiosperm tissues. This also implies no further alteration of Amazon-derived lignin after it is discharged into the ocean and deposited in marine sediments. Lignin in offshore marine sediments thus can provide reliable evidence for the reconstruction of the vegetation cover in the Amazon basin.

The distribution of the degradation state of lignin based on $(Ad/Al)_V$ is shown in Fig. 5D. The strikingly elevated $(Ad/Al)_V$ values in the Amazon Fan are probably caused by longer exposure to oxygen (Blair and Aller, 2012) at the sediment-water interface under low sedimentation rates, corroborating our previous interpretation of the low Λ8 values but intermediate $\delta^{13}C_{TOC}$ values in the Fan sediments (Fig. 6). In the NW sector, there is no obvious decreasing trend of the $(Ad/Al)_V$ values with the distance from the river mouth. Low $(Ad/Al)_V$ values found at shallow nearshore sites far from the Amazon River

mouth (e.g., GeoB16218-3 and GeoB16225-2) may be due to rapid transport and deposition of the material discharged from the Amazon River (Nittrouer et al., 1995). Neither S/V nor C/V ratios decrease with $(Ad/Al)_{V,S}$ in the marine sediments, which would be expected because cinammyl and syringyl pheonls are preferentially degraded compared to vanillyl phenols (Benner et al., 1990; Opsahl and Benner, 1995). Despite the fact that the degradation extent of lignin preserved in marine sediments is slightly higher than that preserved in riverbed sediments, degradation has no major impact on the lignin

composition.

### 5.2.2 Influence of grain size on $OC_{terr}$ deposition in marine sediments

The grain size and Al/Si in the Amazon Fan and SE sectors vary within a rather small range. The grain size and Al/Si relationship in the NW sector is in accordance with the results obtained by Bouchez et al. (2011). The sediments in the NW sector have similar Al/Si ratios as our riverbed sediments (Table 2) which are correlated with grain size. We use grain size

data for the following discussion of the sedimentological control on the distribution pattern of $OC_{terr}$ in the NW sector and refer to the relationship between TOC and Al/Si as observed in the riverbed sediments.

Fine sand sediments were observed at the position closest to the Amazon River mouth (GeoB16209-2) and at site GeoB16225-2, which is far from the Amazon River mouth (about 700 km) but near the coast in a water depth of 34 m. The trend of increasing TOC contents with decreasing grain size (Fig. 7A) parallels the one demonstrated for the riverbed

sediments (Fig. 4A) and in other marine sediments (Keil et al, 1997; Mayer 1994). The lignin content in the organic matter (Λ8) and grain size are not significantly related likely because the $OC_{terr}$ is mixed with marine-derived organic matter in marine environments (Fig. 7B). For example, according to the enriched $\delta^{13}C_{TOC}$ values (mean=-20.4±0.1 ‰), sites GeoB16216-2, GeoB16217-1 and GeoB16223-1, which are the sites located farthest offshore, contain the largest fractions of



marine organic matter, which reduces their Λ8 values to about 0.18 despite their small mean grain size (<14 μm). Except for these three locations, the Λ8 values are higher in finer grained sediments than in sandy sediments. This suggests that in marine sediments, as in the riverbed sediments, sorption of lignin on finer sediment is the dominant control on its distribution.

C/V and (Ad/Al)$_V$ values are not related to the grain size in the NW sector (Fig. 7C, D), which suggests that the influence of grain size on lignin composition and degradation is not as important as in the riverbed sediments. The control on the degradation of lignin on the inner Brazil-French Guiana shelf and slope is probably complex and influenced by many factors, including oxygen exposure time, contribution of material by coastal rivers, and sedimentation rate. Compared with riverbed sediments, offshore sediments also exhibit better preservation of organic matter and selective preservation of lignin in finer

grain size particles, but grain size has limited impact on the lignin composition and degradation status.

In summary, the spatial patterns of lignin content and isotope compositions of organic matter corroborates earlier findings (Geyer et al., 1996; Nittrouer and DeMaster, 1996; Schlünz et al., 1999) that material discharged by the Amazon River is transported northwestward by the North Brazil Current. The modern Amazon Fan area receives more marine organic matter, and petrogenic organic matter is a significant component of OC$_{terr}$ in the Amazon Fan sediments. The similarity of lignin

composition (C/V and S/V) of marine and riverbed sediments suggests that lignin is a reliable tracer reflecting the plant source of terrestrial organic matter in the Amazon basin and can be applied to reconstruct vegetation changes and paleoclimate conditions. Organic matter and lignin content furthermore vary with sediment grain size in the Amazon shelf and slope area and show the same preservation trend, better preservation in finer grained sediments, as in riverbed sediments. However, lignin composition is rather uniform in sediments of different grain sizes.

**6 Conclusions**

In this study, we use TOC content, stable carbon isotopic composition of organic matter, lignin phenol concentrations, sediment grain size and Al/Si ratios (as indicator of grain size) to investigate the characteristics of OC$_{terr}$ in the lowland Amazon basin and its fate on the adjacent continental margin. Depleted δ$^{13}$C$_{TOC}$ of all riverbed sediments prove that there are limited contributions from C4 plants to the OC$_{terr}$ in the lowland Amazon basin. As evidenced by lignin compositions and

stable carbon isotopes of organic matter, the most important plant sources of organic matter in the lowland Amazon basin are non-woody angiosperm C3 plants. There are no distinct regional lignin compositional signatures in the lowland Amazon basin, although the Amazon River receives contributions from tributaries draining different watersheds. Both the bulk organic matter parameters and the lignin compositions were observed to be related to the grain size of the riverbed sediments. Fine inorganic particles in the Amazon River carry more organic matter, preferentially preserving lignin against

degradation. Lignin with higher C/V ratio is inclined to be adsorbed to coarse inorganic particles.

In marine surface sediments, the bulk parameters and lignin compositions indicate that the continental shelf southeast of the Amazon River mouth receives little OC$_{terr}$ from the Amazon River. Most of the OC$_{terr}$ discharged from the Amazon River is

transported by the North Brazil Current to the northwest and deposited on the continental shelf close to the coast. Modern organic matter in the Amazon Fan is composed predominantly of marine-derived organic matter and the terrestrial organic matter undergoes extensive diagenetic alteration before deposition. On the Amazon shelf, the $OC_{terr}$ and lignin are both associated preferentially with fine-grained sediments. Despite long-distance transport in the marine environment, the lignin

composition found in the marine sediments retains its plant source information in accordance with riverbed sediments. Lignin can thus be used to reliably provide assessments on the integrated vegetation cover in the Amazon basin.

**Acknowledgements**

We would like to thank the crews participating in the cruises for providing the samples. We sincerely appreciate the technical support from Jürgen Titschack for grain size measurement. We thank Maria Winterfeld for laboratory assistance.

This study was supported by the Deutsche Forschungsgemeinschaft through the DFG Research Centre/ Cluster of Excellence "The Ocean in the Earth System". S. S. thanks the China Scholarship Council (CSC) and GLOMAR-Bremen International Graduate School for Marine Sciences for additional support. C. M. C. was supported by FAPESP (grant 2012/17517-3) and CAPES (grants 1976/2014 and 564/2015). Sediment sampling in the Amazon River system was funded by FAPESP (grant 2011/06609-1). AOS is supported by CNPq (grant 309223/2014-8). P. A. B. is supported by NSF 1338694.

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



**Table 1. Sample information, results of bulk OC and lignin analyses of riverbed sediment samples from the Amazon basin presented in this study.**

| Sample code | River | Long | Lat | TOC (%) | $\delta^{13}C_{TOC}$ (‰) | Λ8 (mg/100mgOC) | C/V | S/V | P/V | (Ad/Al)v | (Ad/Al)s | Σ8 (mg/10g dry sediment) | Al/Si (weight ratio) |
|---|---|---|---|---|---|---|---|---|---|---|---|---|---|
| MAO02d | Negro | -60.36 | -3.07 | 1.44 | -29.5 | 3.79 | 0.26 | 0.82 | 0.34 | 0.59 | 0.44 | 5.44 | 0.19 |
| MAO02e | Negro | -60.35 | -3.06 | 2.93 | -29.3 | 4.62 | 0.13 | 0.85 | 0.24 | 0.43 | 0.30 | 13.53 | 0.51 |
| MAO02f | Negro | -60.35 | -3.05 | 1.22 | -27.3 | 4.46 | 0.13 | 0.97 | 0.25 | 0.59 | 0.32 | 5.44 | 0.43 |
| MAO1 | Negro | -60.30 | -3.06 | 3.99 | -29.3 | 4.89 | 0.14 | 1.09 | 0.23 | 0.61 | 0.30 | 19.52 | 0.50 |
| MAO03a | Negro | -60.20 | -3.05 | 2.13 | -26.5 | 4.73 | 0.10 | 0.70 | 0.25 | 0.56 | 0.26 | 10.08 | 0.49 |
| MAO03h | Negro | -60.20 | -3.05 | 0.53 | -29.6 | 3.55 | 0.18 | 0.91 | 0.31 | 0.71 | 0.36 | 1.89 | 0.17 |
| MAO4 | Negro | -60.15 | -3.11 | 2.31 | -29.8 | 4.59 | 0.18 | 0.81 | 0.28 | 0.35 | 0.24 | 10.59 | 0.31 |
| MAO11c | Solimões | -60.38 | -3.30 | 0.90 | -29.9 | 3.86 | 0.20 | 1.12 | 0.24 | 0.32 | 0.20 | 3.48 | 0.37 |
| MAO09b | Solimões | -60.29 | -3.27 | 0.28 | n.a. | 3.49 | 0.17 | 1.19 | 0.30 | 0.28 | 0.15 | 0.99 | 0.30 |
| MAO08b | Solimões | -60.21 | -3.30 | 0.59 | -27.9 | 4.97 | 0.21 | 0.99 | 0.25 | 0.33 | 0.28 | 2.93 | 0.37 |
| MAO08a | Solimões | -60.20 | -3.29 | 0.74 | -28.4 | 4.34 | 0.24 | 1.13 | 0.22 | 0.32 | 0.23 | 3.19 | 0.34 |
| MAO05a | Solimões | -60.03 | -3.27 | 0.50 | -26.1 | 4.30 | 0.28 | 1.10 | 0.23 | 0.52 | 0.34 | 2.13 | 0.27 |
| MAO05d | Solimões | -60.02 | -3.29 | 0.57 | -28.4 | 4.71 | 0.1 | 0.94 | 0.17 | 0.35 | 0.26 | 2.69 | 0.33 |
| MAO13c | Solimões | -59.88 | -3.20 | 0.77 | -28.4 | 6.44 | 0.17 | 1.06 | 0.24 | 0.28 | 0.19 | 4.94 | 0.37 |
| MAO23a | Madeira | -59.08 | -3.68 | 0.52 | -28.4 | 4.42 | 0.14 | 0.96 | 0.21 | 0.29 | 0.23 | 2.30 | 0.37 |
| MAO25e | Madeira | -58.91 | -3.52 | 0.47 | -27.8 | 3.29 | 0.21 | 1.21 | 0.26 | 0.35 | 0.22 | 1.55 | 0.31 |
| MAO28d | Madeira | -58.80 | -3.44 | 0.52 | -27.9 | 3.10 | 0.24 | 1.17 | 0.25 | 0.39 | 0.25 | 1.61 | 0.23 |
| MAO23c | Madeira | -59.08 | -3.67 | 0.16 | n.a. | 1.69 | 0.29 | 1.04 | 0.44 | 0.49 | 0.40 | 0.28 | 0.17 |
| MAO25d | Madeira | -58.91 | -3.53 | 0.14 | n.a. | 0.73 | 0.36 | 1.07 | 0.63 | 0.67 | 0.57 | 0.10 | 0.17 |
| MAO25b | Madeira | -58.91 | -3.53 | 0.35 | n.a. | 2.42 | 0.23 | 1.16 | 0.48 | 0.71 | 0.38 | 0.84 | 0.31 |
| MAO15e | Amazon | -59.38 | -3.19 | 0.25 | n.a. | 2.84 | 0.20 | 1.14 | 0.18 | 0.31 | 0.21 | 0.70 | 0.24 |
| MAO15a | Amazon | -59.38 | -3.15 | 0.80 | -27.9 | 4.01 | 0.1 | 0.91 | 0.20 | 0.40 | 0.24 | 3.21 | 0.36 |
| MAO17 | Amazon | -59.29 | -3.15 | 0.59 | -28.9 | 3.51 | 0.14 | 1.16 | 0.34 | 0.37 | 0.20 | 2.06 | 0.26 |
| MAO19 | Amazon | 59.13 | -3.18 | 1.00 | -29.4 | 4.22 | 0.14 | 0.92 | 0.23 | 0.48 | 0.32 | 4.22 | 0.37 |
| MAO21f | Amazon | -59.03 | -3.23 | 0.99 | -28.7 | 5.04 | 0.12 | 0.96 | 0.26 | 0.29 | 0.21 | 5.01 | 0.38 |
| MAO36 | Amazon | -58.62 | -3.25 | 1.44 | -28.1 | 4.76 | 0.21 | 0.84 | 0.24 | 0.45 | 0.26 | 6.85 | 0.31 |
| OB1 | Amazon | -55.56 | -1.89 | 0.61 | -28.4 | 3.79 | 0.21 | 1.10 | 0.30 | 0.48 | 0.28 | 2.33 | 0.32 |
| MC11 | Amazon | -51.09 | -0.06 | 0.13 | n.a. | 0.75 | 0.47 | 1.01 | 0.64 | 0.61 | 0.56 | 0.10 | 0.11 |
| MC12-2 | Amazon | -51.05 | -0.08 | 1.19 | -28.3 | 9.27 | 0.11 | 0.88 | 0.17 | 0.28 | 0.25 | 11.05 | 0.37 |



| | | | | | | | | | | | | |
|---|---|---|---|---|---|---|---|---|---|---|---|---|
| MC12-1 | Amazon | -51.05 | -0.08 | 1.01 | -28.1 | 7.20 | 0.13 | 1.05 | 0.19 | 0.42 | 0.29 | 7.29 | 0.36 |
| MC8 | Amazon | -50.66 | -0.13 | 0.62 | -27.5 | 5.30 | 0.13 | 1.20 | 0.37 | 0.30 | 0.19 | 3.27 | 0.31 |
| MC1 | Amazon | -50.65 | -0.12 | 0.60 | -28.0 | 3.83 | 0.16 | 1.51 | 0.35 | 0.35 | 0.15 | 2.29 | 0.30 |
| MC2 | Amazon | -50.64 | -0.13 | 0.40 | -28.0 | 3.82 | 0.15 | 1.14 | 0.20 | 0.36 | 0.22 | 1.53 | 0.26 |
| MC3 | Amazon | -50.62 | -0.15 | 0.26 | n.a. | 6.36 | 0.15 | 1.17 | 0.19 | 0.26 | 0.22 | 1.67 | 0.18 |
| MC4 | Amazon | -50.61 | -0.17 | 0.72 | -28.0 | 5.06 | 0.16 | 1.14 | 0.20 | 0.27 | 0.19 | 3.64 | 0.30 |
| MC7 | Amazon | -50.59 | -0.21 | 0.37 | n.a. | 3.51 | 0.13 | 1.08 | 0.29 | 0.38 | 0.22 | 1.29 | 0.27 |
| MC5 | Amazon | -50.58 | -0.20 | 1.29 | -28.0 | 7.11 | 0.11 | 1.01 | 0.18 | 0.30 | 0.31 | 9.18 | 0.23 |
| MC6 | Amazon | -50.56 | -0.19 | 0.80 | -27.8 | 5.13 | 0.21 | 1.31 | 0.33 | 0.57 | 0.25 | 4.10 | 0.29 |
| MC10 | Amazon | -50.09 | -0.05 | 0.76 | -28.4 | 7.18 | 0.11 | 1.22 | 0.35 | 0.30 | 0.23 | 5.49 | 0.26 |
| XA14L | Xingu | -52.69 | -3.88 | 3.37 | -29.7 | 5.33 | 0.16 | 0.98 | 0.20 | 0.55 | 0.36 | 17.97 | 0.29 |
| XA30 | Xingu | -52.24 | -1.69 | 0.83 | -27.9 | 3.30 | 0.16 | 1.23 | 0.28 | 0.31 | 0.18 | 2.73 | 0.33 |
| XA36 | Xingu | -52.13 | -2.22 | 3.24 | -29.8 | 3.33 | 0.08 | 0.90 | 0.24 | 0.43 | 0.26 | 10.81 | 0.55 |
| XA25 | Xingu | -51.97 | -2.64 | 3.82 | -29.4 | 3.90 | 0.19 | 1.12 | 0.25 | 0.42 | 0.27 | 14.90 | 0.54 |
| XA38 | Xingu | -52.02 | -2.47 | 3.52 | -29.6 | 4.47 | 0.15 | 1.05 | 0.25 | 0.55 | 0.28 | 15.73 | 0.49 |
| XA31 | Xingu | -52.25 | -1.79 | 1.11 | -28.4 | 6.91 | 0.19 | 0.91 | 0.19 | 0.34 | 0.26 | 7.70 | 0.30 |
| XA34 | Xingu | -52.26 | -1.79 | 0.52 | -28.3 | 6.88 | 0.19 | 0.75 | 0.40 | 0.38 | 0.27 | 3.55 | 0.14 |
| XA35 | Xingu | -52.19 | -2.04 | 3.07 | -29.8 | 5.22 | 0.16 | 1.03 | 0.36 | 0.53 | 0.37 | 16.04 | 0.56 |

**n.a.: not available**



**Table 2. Sample information, results of bulk OC and lignin analyses of marine surface sediment samples from the Amazon continental margin presented in this study.**

| Sample code | Long | Lat | TOC (%) | $\delta^{13}C_{TOC}$ (‰) | Λ8 (mg/100mg OC) | S/V | C/V | (Ad/Al)v | (Ad/Al)s | Σ8 (mg/10g sediment) | Al/Si (weigt ratio) | Mean grain size (μm) |
|---|---|---|---|---|---|---|---|---|---|---|---|---|
| The Amazon Fan sector (Fan) | | | | | | | | | | | | |
| BC14B | -48.35 | 4.04 | 0.46 | -23.0 | 0.09 | 0.81 | 0.15 | 0.91 | 0.81 | 0.04 | 0.42 | 5.2 |
| BC17B | -48.54 | 3.96 | 0.44 | -21.8 | 0.14 | 1.33 | 0.13 | 0.99 | 0.65 | 0.06 | 0.42 | 6.2 |
| BC24B | -48.89 | 3.80 | 0.49 | -22.2 | 0.22 | 1.62 | 0.18 | 0.88 | 0.49 | 0.11 | 0.40 | 14.5 |
| BC3C | -48.61 | 4.46 | 0.60 | -21.9 | 0.19 | 1.33 | 0.35 | 0.65 | 0.71 | 0.11 | 0.45 | 3.9 |
| BC44C | -48.17 | 3.78 | 0.44 | -21.8 | 0.10 | 0.59 | 0.11 | 0.75 | 0.70 | 0.04 | 0.41 | 4.7 |
| BC50C | -47.31 | 3.65 | 0.57 | -22.3 | 0.09 | 0.78 | 0.22 | 0.87 | 0.80 | 0.05 | 0.45 | 3.8 |
| BC55C | -47.64 | 3.07 | 0.46 | -21.7 | 0.13 | 0.76 | 0.14 | 0.81 | 0.70 | 0.06 | 0.40 | 8.2 |
| BC61C | -47.74 | 2.85 | 0.98 | -24.5 | 0.57 | 1.28 | 0.24 | 0.92 | 0.61 | 0.37 | 0.46 | 3.2 |
| BC71C | -46.25 | 3.39 | 0.50 | -21.4 | 0.06 | 1.28 | 0.35 | 0.70 | 0.77 | 0.03 | 0.46 | n.a. |
| MC12A | -48.34 | 4.04 | 0.67 | -22.6 | 0.13 | 1.61 | 0.27 | 0.56 | 0.78 | 0.09 | 0.43 | 5.1 |
| MC6A | -48.62 | 4.46 | 0.77 | -22.2 | 0.06 | 1.04 | 0.26 | 0.92 | 0.76 | 0.05 | 0.45 | 3.5 |
| The southeast sector (SE) | | | | | | | | | | | | |
| BC75C | -45.35 | 1.68 | 0.31 | n.a. | 0.04 | 0.81 | 0.25 | 0.86 | 0.92 | 0.01 | 0.40 | n.a. |
| BC80C | -44.35 | 0.66 | 0.21 | n.a. | 0.10 | 0.80 | 0.11 | 0.49 | 0.42 | 0.02 | 0.38 | 4.8 |
| BC82C | -44.21 | 0.34 | 0.35 | -21.6 | 0.05 | 0.91 | 0.31 | 0.80 | 0.77 | 0.02 | 0.40 | 4.4 |
| BC90B | -42.74 | -1.03 | 0.43 | n.a. | 0.08 | 1.25 | 0.20 | 0.76 | 0.48 | 0.04 | 0.40 | 6.0 |
| GeoB16202-1 | -41.59 | -1.91 | 0.47 | -18.6 | 0.06 | 0.86 | 0.27 | 0.50 | 0.64 | 0.05 | 0.37 | 9.3 |
| GeoB16203-2 | -41.72 | -2.04 | 0.75 | -19.1 | 0.08 | 0.98 | 0.43 | 0.51 | 0.67 | 0.07 | 0.41 | 8.3 |
| GeoB16204-1 | -42.34 | -2.00 | 0.91 | -19.6 | 0.09 | 1.05 | 0.43 | 0.52 | 0.68 | 0.10 | 0.39 | 11.8 |
| GeoB16205-3 | -43.10 | -1.35 | 0.50 | -19.7 | 0.17 | 0.70 | 0.23 | 0.54 | 0.53 | 0.12 | 0.40 | 10.9 |
| GeoB16206-2 | -43.02 | -1.58 | 0.55 | -18.9 | 0.07 | 0.87 | 0.43 | 0.59 | 0.85 | 0.05 | 0.39 | 9.1 |
| The northwest sector (NW) | | | | | | | | | | | | |
| MC33A | -49.79 | 3.23 | 0.70 | -26.7 | 2.01 | 1.23 | 0.16 | 0.73 | 0.54 | 1.49 | 0.46 | 5.0 |
| GeoB16209-2 | -49.37 | 2.83 | 0.11 | -23.2 | 0.19 | 0.83 | 0.20 | 0.64 | 0.97 | 0.03 | 0.15 | 88.4 |
| GeoB16210-2 | -49.36 | 2.87 | 0.11 | -24.1 | 0.39 | 1.06 | 0.29 | 0.50 | 0.48 | 0.04 | 0.17 | 48.7 |





| | | | | | | | | | | | | |
|---|---|---|---|---|---|---|---|---|---|---|---|---|
| GeoB16211-2 | -49.35 | 2.88 | 0.18 | -24.1 | 1.08 | 0.84 | 0.12 | 0.46 | 0.35 | 0.22 | 0.22 | 41.9 |
| GeoB16212-2 | -49.39 | 3.10 | 0.73 | -25.7 | 1.55 | 1.01 | 0.16 | 0.51 | 0.35 | 1.05 | 0.46 | 5.7 |
| GeoB16216-2 | -51.26 | 6.24 | 0.79 | -20.4 | 0.18 | 1.17 | 0.31 | 0.66 | 0.90 | 0.17 | 0.41 | 5.7 |
| GeoB16217-1 | -51.29 | 6.07 | 0.50 | -20.3 | 0.18 | 1.10 | 0.25 | 1.16 | 0.76 | 0.10 | 0.30 | 13.1 |
| GeoB16218-3 | -51.52 | 4.77 | 0.76 | -23.7 | 1.18 | 0.99 | 0.16 | 0.47 | 0.32 | 0.95 | 0.47 | 4.7 |
| GeoB16223-1 | -52.12 | 6.63 | 0.79 | -20.5 | 0.19 | 1.11 | 0.29 | 0.65 | 0.86 | 0.17 | 0.38 | 6.1 |
| GeoB16225-2 | -52.86 | 5.67 | 0.27 | -21.7 | 1.05 | 0.63 | 0.10 | 0.37 | 0.32 | 0.33 | 0.14 | 92.9 |

**n.a.: not available**



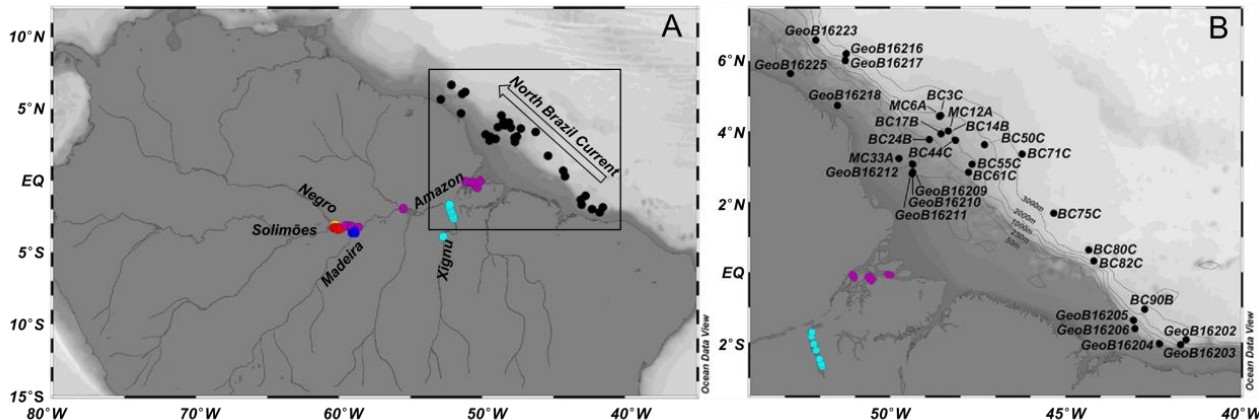

**Figure 1: A) Map of the Amazon basin with the sample locations in individual tributaries and offshore area indicated by colored dots (red dots=Solimões, yellow dots=Negro, blue dots=Madeira, aqua dots=Xingu, violet dots=Amazon mainstream, black dots=offshore), the black rectangle indicates the area of map B and B) Map of the Amazon continental margin with the sample locations indicated by black dots. Maps were created using Ocean Data View 4.7.8 (Schlitzer, 2016).**

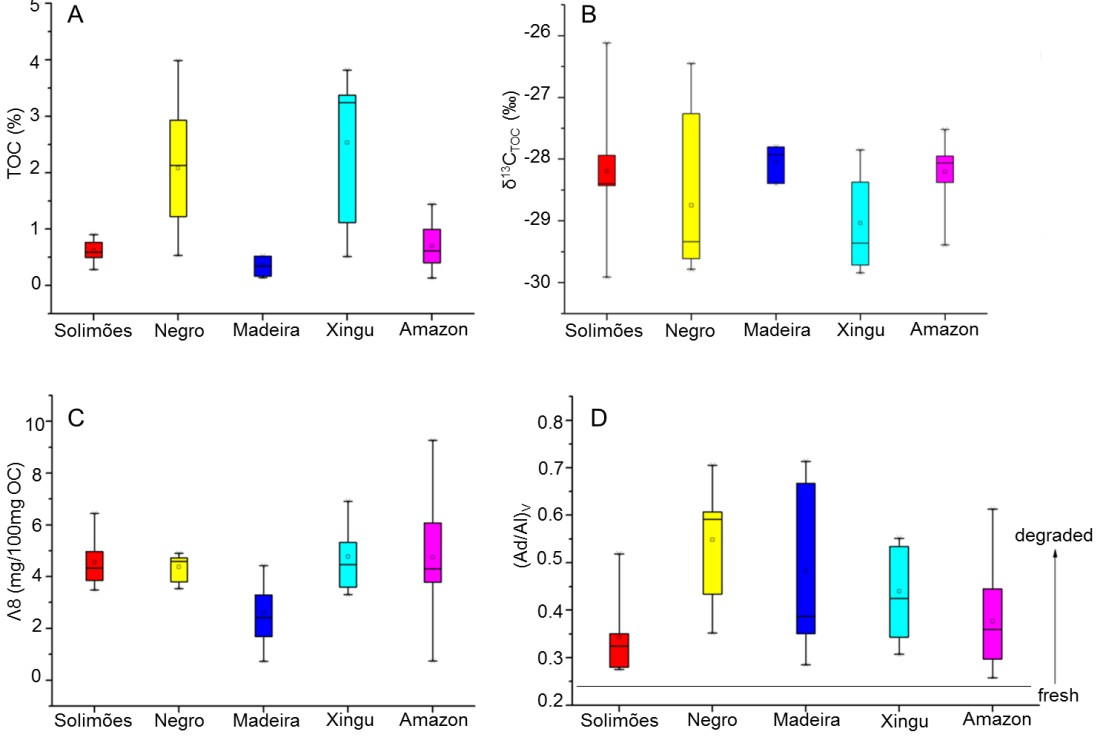

**Figure 2: A) Total organic carbon (TOC) content, B) stable carbon isotopic composition of total organic carbon ($\delta^{13}C_{TOC}$), C) carbon-normalized lignin content ($\Lambda 8$) and D) degradation index of lignin $(Ad/Al)_V$ in different tributaries and in the Amazon mainstream. In D, $(Ad/Al)_V \leq 0.24$ suggests fresh vascular plant tissues, whereas $(Ad/Al)_V > 0.24$ reveals degraded plant material. See Table 1 and Fig. 1 for the location of the samples.**





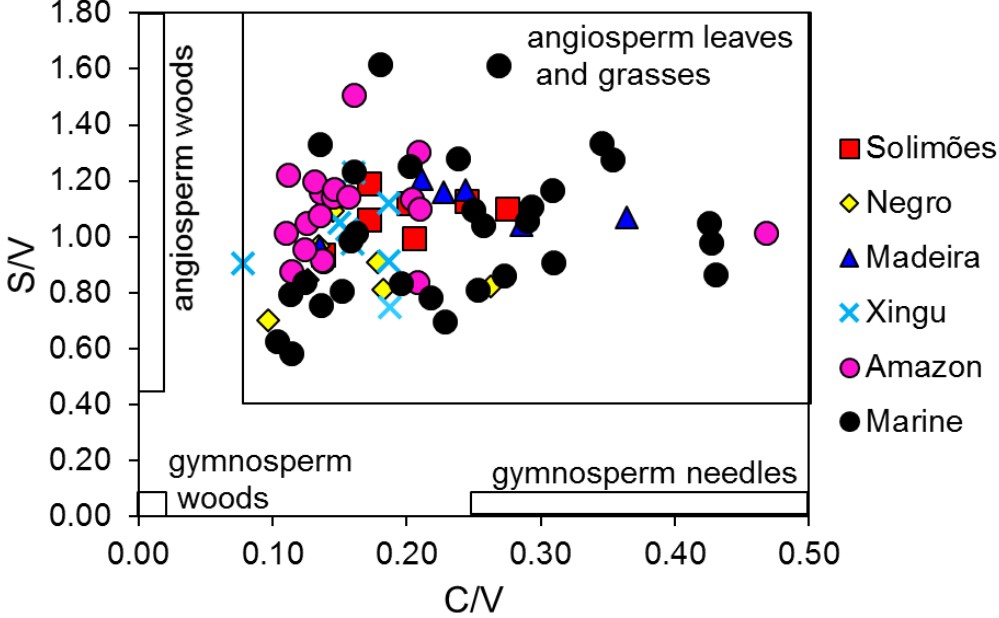

**Figure 3: Syringyl:vanillyl (S/V) vs. cinnamyl:vanillyl (C/V) ratios of lignin from Amazon basin river bed sediments and marine surface sediments from the adjacent continental margin. The black boxes show typical ranges for different vascular plant tissues from the Amazon basin (Hedges et al., 1986, Goñi et al.,1998). See Tables 1 and 2, and Fig. 1 for the location of the samples.**

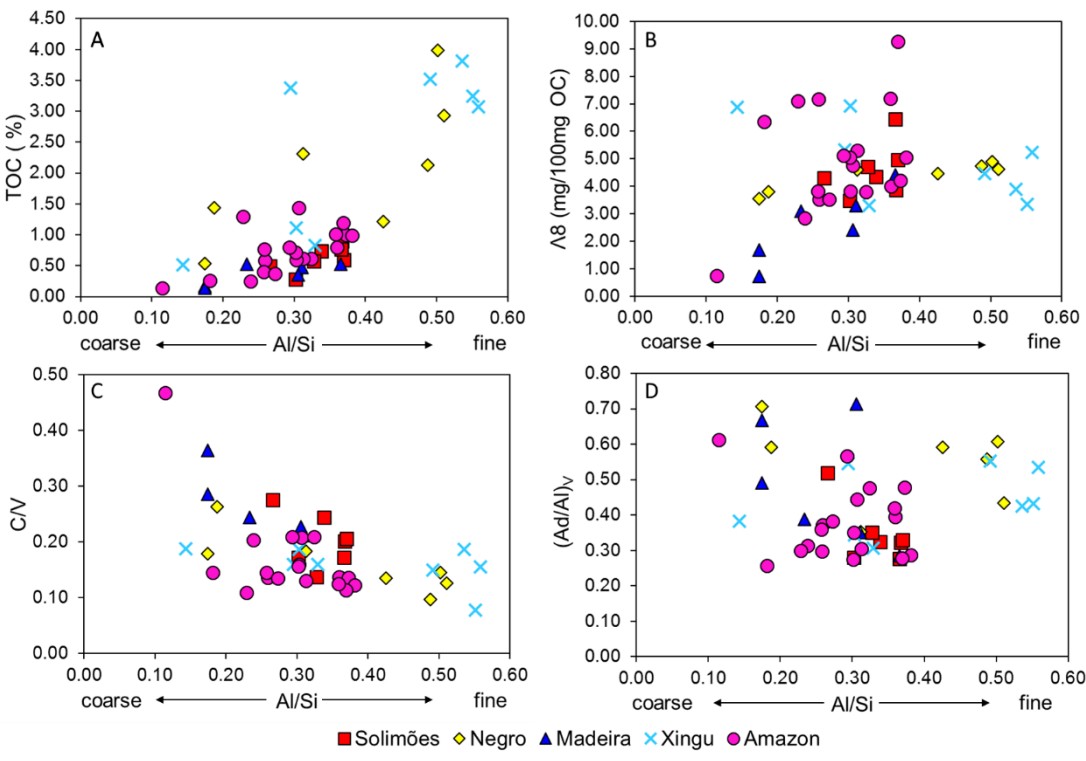



**Figure 4: A) Total organic carbon (TOC) content vs. Al/Si, B) carbon-normalized lignin content (Λ8) vs. Al/Si, C) cinnamyl:vanillyl (C/V) ratio vs. Al/Si, and D) degradation index of lignin ((Ad/Al)$_V$) vs. Al/Si from Amazon basin riverbed sediments. See Table 1 and Fig. 1 for the location of the samples.**

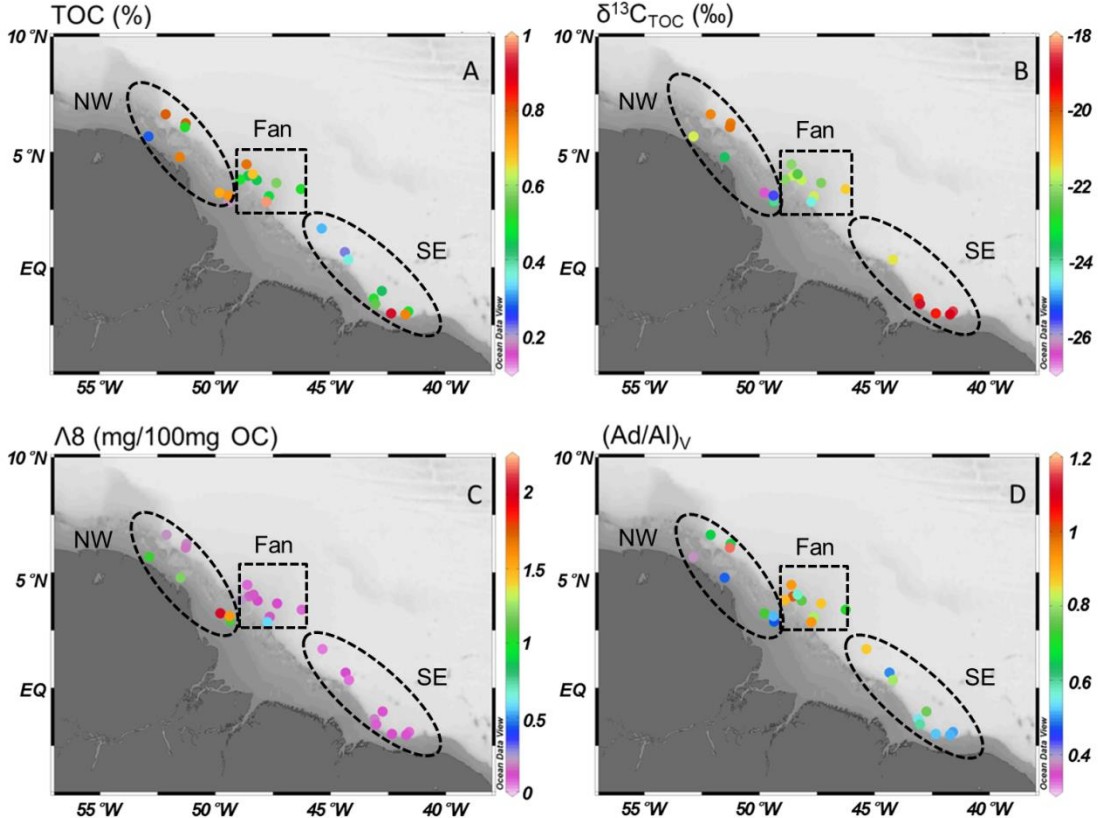

5   **Figure 5: Spatial distribution of A) total organic carbon (TOC) content, B) stable carbon isotopic composition of total organic carbon (δ$^{13}$C$_{TOC}$), C) carbon-normalized lignin content (Λ8), and D) degradation index of lignin ((Ad/Al)$_V$) in marine surface sediment samples from the Amazon continental margin. The dashed ellipses and the rectangle represent the northwest sector (NW), the Amazon Fan sector (Fan) and southeast sector (SE).**



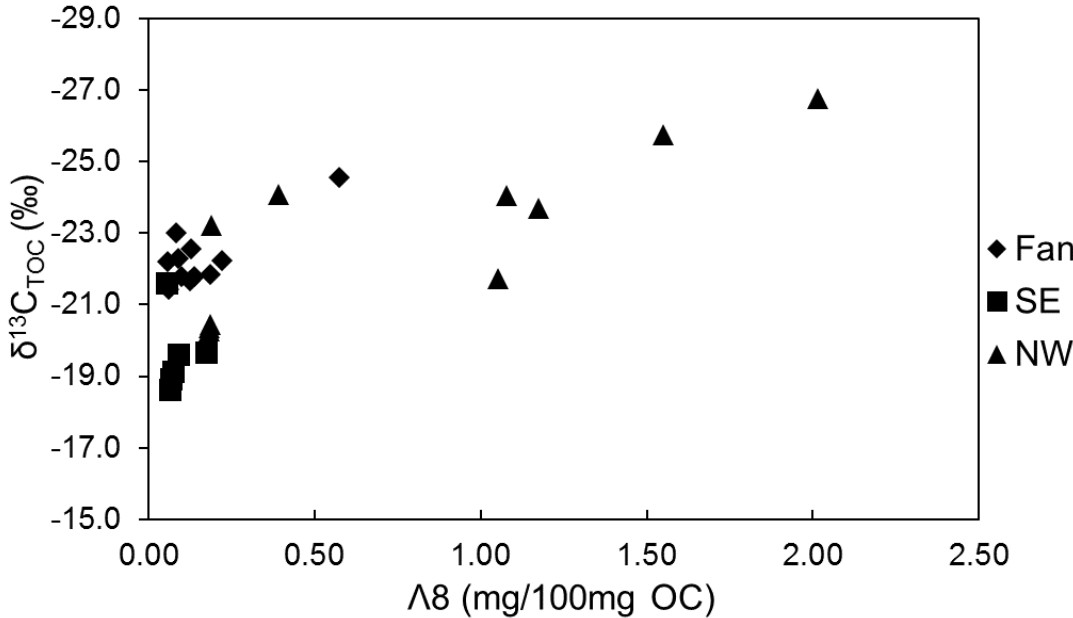

**Figure 6: Stable carbon isotopic composition of total organic carbon (δ¹³C_TOC) vs. carbon-normalized lignin content (Λ8) for marine surface sediment samples from the Amazon continental margin.**





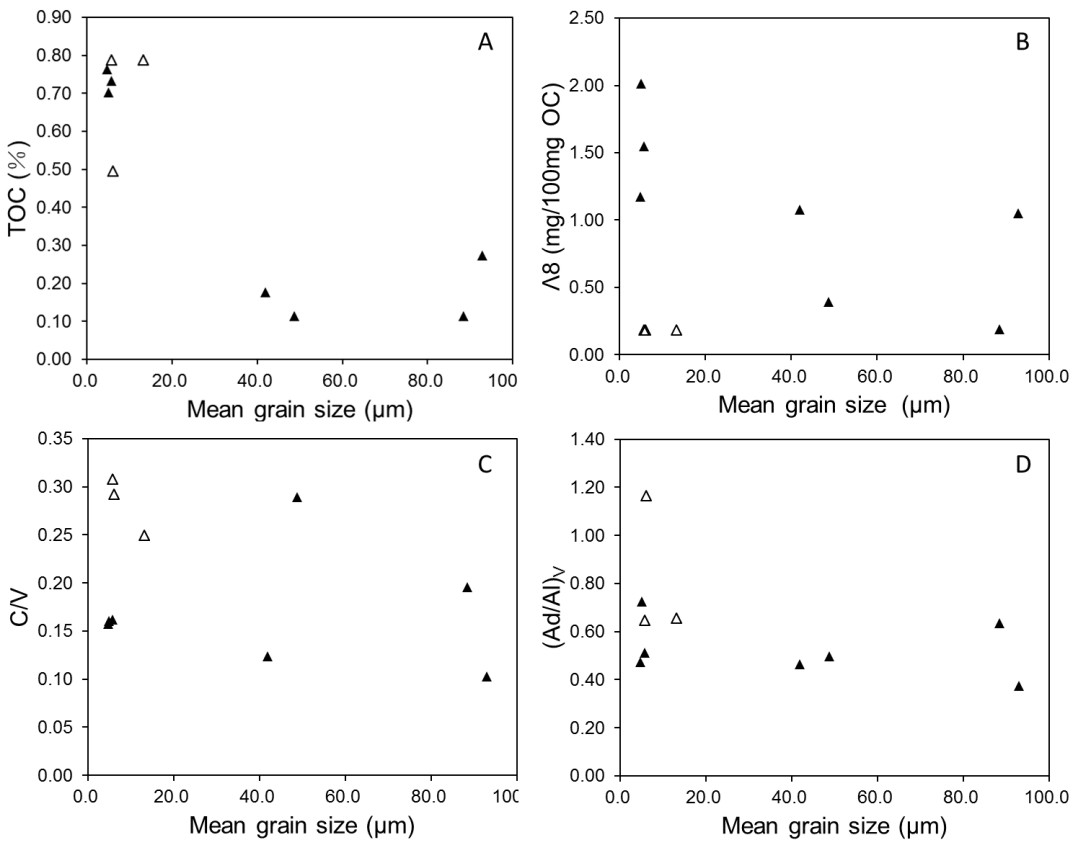

**Figure 7: A), B), C) and D) indicate total organic carbon (TOC) content, carbon-normalized lignin content (Λ8), cinnamyl:vanillyl (C/V) ratio and lignin degradation index ((Ad/Al)$_V$) for marine surface sediment samples from the NW sector vs. mean grain size, respectively. Empty triangles represent the three deepest sites (>2000 m) far from the coast, and filled triangles represent the other sites in the NW sector with water depth shallower than 100 m. See Table 2 and Fig. 5 for the location of the NW sector.**