# Peer review of "Origin and processing of terrestrial organic carbon in the Amazon system: lignin phenols in river, shelf and fan sediments"

_Biogeosciences, 2016_

## Referee Comment (RC1) · X. Feng (Referee) · 14 Dec 2016

The transport and deposition of terrestrial organic matter in the Amazon River system is an important topic as the Amazon is a globally important exporter of terrestrial carbon into the ocean. As such, lignin distribution and transport in the Amazon River has received much attention in the past but connections or direct comparisons between the fluvial system versus marine archives are rare. To fill in this gap, this paper examines the distribution of lignin phenols together with bulk OC and 13C in riverbed sediments from the Amazon mainstream and main tributaries as well as marine surface sediments from the Amazon shelf and fan. Overall, the paper is clearly written with large amounts of original data. There are some interesting findings that are unique compared with

other fluvial systems. But as I read the paper for the first time, I found myself a bit overwhelmed by the detailed description of parameters. I understand that this is partly due to the large volume of data presented in the paper. More emphases on the (statistically significant) trend or pattern (rather than the actual values) of parameters may help to organize the paper in a more digestible fashion. Also, I would expect more discussions on the implications or comparison with previous Amazonian lignin work or other fluvial systems.

Detailed comments:

Abstract Line 19: Results from this study somehow contrasts with those from some other marine environments. For instance, lignin phenols in the Washington continental shelf and slope (affected by the Columbia River) are concentrated in the sand-size sediments due to the export of plant debris (e.g., Keil et al. 1998 GCA 62, 1347-1364). Tesi et al. (2016 JGR) has also found that lignin distribution in the Siberian Shelf coincides with the distribution of coarse particles enriched in plant debris. This study, however, reports that lignin is preferentially preserved in fine particles. What characteristics of the Amazon basin make lignin distribution in particles different from that in the Washington and Siberian Shelves? Does it have anything to do with the narrow range of Al/Si ratios, indicating relatively coarse particle sizes of the studied sediments? This should be discussed in the paper.

Page 7, Line 9: why is Xingu different from the other tributaries and mainstem?

Page 8, Line 9: lignin increases with distance from the river mouth—this is interesting. Is it statistically significant? If so, why?

Page 11, Line 1: here and the discussion below mentions that the processing of terrestrial OC is minimal in the Amazon. This contradicts with the findings by Ward et al. (2013 Nature Geoscience) that lignin is being fast degraded within the Amazon River. How do you reconcile the discrepancy? This should also be discussed in the paper.

Page 11, Line 32: it is commonly thought that OC in fine particles are more degraded. Your observation is different. Feng et al. (2015 JGR) has found that lignin phenols in the forest O horizons of the upper Amazon basin exhibit very high Ad/Al values. Does this have a play in your observation? This may also explain the constant ratios of C/V and S/V (unaffected by degradation) versus increasing Ad/Al ratios?

Page 13, Line 10: "Spatial variation" of the degradation state of lignin. . ..

References: - Feng et al. 2016 JGR has recently investigated and compiled lignin phenol distributions in the soil-river (POM & DOM)-marine sediment continuum of the Madre de Dois-Amazon system. This paper may be a good comparison to your data.

Xiaojuan Feng

---

## Referee Comment (RC2) · R. Sparkes (Referee) · 3 Jan 2017

Sun et al have produced a robust dataset investigating the distribution of organic carbon, especially lignin phenols, in the Amazon River and Fan system. Through a combination of techniques they have shown that degraded lignin is transported through the system and buried hundreds of kilometres from the river mouth, preferentially attached to fine sediment particles.

The manuscript is well written and presents a good data set. I recommend that it is published subject to the following few revisions/suggestions.

Minor comments:

[Figure]

Page 9 Line 10. The phrase "reflects the characteristics of the tributaries (Fig 2A)" was a bit confusing. After reading through the following section it seems that dilution of carbon by mineral sediment is the main reason for changes in TOC. Authors could consider being more explicit about this at the start of the section.

Page 9 Line 28. A good relation is claimed between $\Lambda 8$ and $\Sigma 8$. Authors could consider referring to a figure or providing the p value.

Page 12 Line 15. Authors could use a simple mixing model to estimate the proportion of marine and terrestrial carbon in their offshore sediments.

Page 12 Line 25. $\Lambda 8$ and d13C are positively correlated. Authors could show this using p values or a figure.

Page 13 Line 28. Sample GeoB16225-2 is 30 km offshore, 30 km from a small local river, and in only 34 m water depth with a fine sand sedimentology. Can the authors be confident that this sample reflects material from the Amazon and not local input?

Typographical changes:

Page 6 Line 13. "were". The rest of this paragraph is in the present tense

Page 7 Line 15. "varied". The rest of this paragraph is in the present tense

Page 9 Line 19. "‰ )". There is an extra space

Page 9 Line 20. "Bouchez et al. ,2014". There is a space in the wrong location

Page 12 Line 24. "reaching very low $\Lambda 8$ value". It is unclear whether the authors mean to use the singular or plural form "reaching a very low $\Lambda 8$ value" / "reaching very low $\Lambda 8$ values"

---

## Author Comment (AC1) · 15 Mar 2017

Dear referees and editor,

We thank you for the time and effort in reviewing our manuscript. We are pleased to hear that you have found our work interesting, and we are also grateful for the issues you pointed out to help us improve the quality of our work.

Motivated by your comments, we have restructured our manuscript and tried to address all the issues you mentioned. The responses to your comments and the corresponding changes in the manuscript are listed in the following:

**Comments from X. Feng:**

Abstract Line 19:

Results from this study somehow contrasts with those from some other marine environments. For instance, lignin phenols in the Washington continental shelf and slope (affected by the Columbia River) are concentrated in the sand-size sediments due to the export of plant debris (e.g., Keil et al. 1998 GCA 62, 1347-1364). Tesi et al. (2016 JGR) has also found that lignin distribution in the Siberian Shelf coincides with the distribution of coarse particles enriched in plant debris. This study, however, reports that lignin is preferentially preserved in fine particles. What characteristics of the Amazon basin make lignin distribution in particles different from that in the Washington and Siberian Shelves? Does it have anything to do with the narrow range of Al/Si ratios, indicating relatively coarse particle sizes of the studied sediments? This should be discussed in the paper.

**Response:**

We have noticed that our conclusion that the lignin is preferentially preserved in fine particles is in contrast to some previous studies. We do not think this is due to the narrow range of Al/Si ratios. Our Al/Si ratios are between 0.11 and 0.56, which represent grain size (reported as D90) from 10 µm to 330 µm, according to Bouchez et al. (2011, GGG, 12). Even the narrow Al/Si ranges in the Solimões and Madeira rivers represent mean grain sizes varying from 41 µm to 191µm. This range furthermore includes both of the size fractions investigated in earlier studies (i.e., the coarse fraction> 63 µm, the fine fraction<63 µm) (Hedges et al. 1986 Limnol Oceanogr. 31).

In our opinion, the discrepancy likely derives from the different methods of characterizing the grain size distribution of lignin on one hand, and on the other hand might be the result of differences in climatic settings which influences the sediment composition.

In the studies of Keil et al. (1998 GCA 62, 1347-1364) and Tesi et al. (2016 JGR), the sediments were separated into different fractions based on grain size and density. Our samples, in contrast, were not separated in this way. Our sediments were directly ground with the plant debris and mineral particles homogenized. The sediments were thus characterized and discussed as a whole. If we compare our results to those previous studies that like us used mean grain size to explore the effect of grain size distribution on lignin preservation, our results are not in contrast to them. For instance, in the study of Wu et al. (2013 JGR) on the organic matter in surficial sediments of the East China Sea, lignin is enriched in fine-grained sediments, suggesting important contributions form soils. Schmidt et al. (2010 Mar. Chem.) have observed that lignin phenols are preferentially associated with the silt fraction on the Iberian margin.

Alternatively, the discrepancy pointed out by Y. Feng could arise from the distinct climatic settings of the study areas. Our study area in the tropics may be characterized by very efficient degradation

of large plant debris, which is typically enriched in the low density fractions reported to contain high amounts of lignin in the studies by Keil et al. and Tesi et al. Those studies were carried out in higher latitude settings (Washington margin and Laptev Sea), where degradation on land is likely less efficient than in the tropical areas drained by the Amazon River. The particulate organic matter in the riverbed sediments of the lower Amazon basin derives most likely from soils, and plant debris contributes very insignificantly to the sedimentary organic matter. Therefore, it is conceivable that in their studies, a different overall trend in lignin concentrations versus bulk grain size exists. These are the reasons why our results are different from those of Keil et al. (1998 GCA 62, 1347-1364) and Tesi et al. (2016 JGR).

**Changes in manuscript:**
This point is now discussed in the manuscript in the third paragraph in section 5.1.3.

**Comments from X. Feng:**
Page 7, Line 9: why is Xingu different from the other tributaries and mainstem?

**Response:**
Unlike the other tributaries and the Amazon mainstream, the $\Sigma 8$ and $\Lambda 8$ of the Xingu River are not correlated ($r^2$=0.0239). The Xingu River is a clear water river and has higher levels of in situ primary production relative to the turbid Amazon River. The phytoplankton-derived organic matter will settle out of the water column and dilute the lignin derived compounds. This will decrease the carbon-normalized lignin yields ($\Lambda 8$) but does not influence the sediment-normalized parameter $\Sigma 8$.

**Changes in manuscript:**
The ($r^2$=0.0239) of the relation between $\Lambda 8$ and $\Sigma 8$ of the Xingu River is added and the environmental explanation is provided at the second line in section 5.1.2.

**Comments from X. Feng:**
Page 8, Line 9: lignin increases with distance from the river mouth. This is interesting. Is it statistically significant? If so, why?

**Response:**
The slight increasing $\Lambda 8$ with distance from the Amazon river mouth was observed only in the SE sector. It is commonly thought that the lignin content should decrease with the distance from the river mouth, because the terrestrial organic matter is diluted by the marine organic matter during the transport. However, in the SE sector we observe the opposite trend. So, we conclude that the lignin in the SE sector derived from the small rivers in the southeast area. This inference is consistent with the evidence that the distribution of the Amazon River plume and the associated sedimentation are affected by the North Brazil Current, which flows northwestward. Because our focus is on the organic carbon derived from the Amazon River, so we did not emphasize details about the SE sector.

**Changes in manuscript:**

The explanation outlined above is now included in the second paragraph in section 5.2.1.

**Comments from X. Feng:**

Page 11, Line 1: here and the discussion below mentions that the processing of terrestrial OC is minimal in the Amazon. This contradicts with the findings by Ward et al. (2013 Nature Geoscience) that lignin is being fast degraded within the Amazon River. How do you reconcile the discrepancy? This should also be discussed in the paper.

**Response:**

In the study of Ward et al. (2013 Nature), dissolved and particulate lignin and phenolic concentrations from Amazon waters were measured before and after incubation experiments. Their conclusion of a rapid degradation is based on the finding of a post-incubation decrease in concentrations of phenolic compounds. The lignin measured in the study of Ward et al. (2013) exists in water column as free particulate and dissolved phase, which is exposed to favourable conditions for degradation. However, our study focuses on the lignin associated with mineral particles,which are deposited and protected from degradation. That is the reason for the contradiction.

**Changes in manuscript:**

The point is discussed at the end in section 5.1.2.

**Comments from X. Feng:**

Page 11, Line 32: it is commonly thought that OC in fine particles are more degraded. Your observation is different. Feng et al. (2015 JGR) has found that lignin phenols in the forest O horizons of the upper Amazon basin exhibit very high Ad/Al values. Does this have a play in your observation? This may also explain the constant ratios of C/V and S/V (unaffected by degradation) versus increasing Ad/Al ratios?

**Response:**

The apparent discrepancy between our finding and the common expectation about the correlation between Ad/Al values and grain size that is pointed out here has a similar origin as the issue discussed above, i.e. the explanation for why lignin is enriched in fine-grained particles in our study but more abundant in coarser fraction in some other studies. It probably depends on whether the sediments are studied as different grain size fractions or as bulk sample. In the study of Schmidt et al. (2010 Mar. Chem.), who like us studied bulk sediments, the Ad/Al values are positively related to mean grain size, which indicates the lignin is less degraded in finer grained particles. This is consistent with our observation.

Moreover, the distribution of Ad/Al ratios within grain size classes is related to the content of plant debris. If plant debris accounts for a large proportion in the sediment, the coarse fraction will show very low Ad/Al values, because of the existence of fresh plant tissue. If the contribution of plant debris is poor, the Ad/Al values will follow a trend as we observed with better preserved lignin in finer grained particles.

The finding of Feng et al. (2016 JGR) provides another possible explanation that the lignin associated with fine particles is more likely from the surface mineral soils (lower Ad/Al values)

and the lignin associated with coarse particles derives from the forest O horizons (higher Ad/Al values). However, their setting in high altitude in Andean area is different from the lowland basin. The Andean signal might not dominate the lowland tributaries. So we refrained from including a speculation based on the observation of Feng et al. (2016 JGR) in the revised manuscript.

In our study, we have observed that the C/V ratios in the the Madeira River, the Solimões River and the Negro River decrease with decreasing grain size. Our C/V ratios and $(Ad/Al)_V$ ratios are positively related, while it is commonly thought that C/V ratios decrease with increasing Ad/Al ratios, because cinnamyl phenols are more easily degraded compared to vanillyl phenols. The S/V ratios are basically constant versus Ad/Al ratios. This unexpected correlation between lignin composition and the Ad/Al ratios probably means S/V and C/V ratios are not influenced by degradation as indicated by Feng et al. (2016 JGR).

**Changes in manuscript:**
The explanation is added in the fourth paragraph in section 5.1.3.

**Comments from X. Feng:**
Page 13, Line 10: "Spatial variation" of the degradation state of lignin…
References: -Feng et al. 2016 JGR has recently investigated and compiled lignin phenol distributions in the soil-river (POM & DOM) –marine sediment continuum of the Madre de DOIS-Amazon system. This paper may be a good comparison to your data.

**Response:**
The sampling locations of marine samples in Feng et al. (2016 JGR) are further away from our samples in the Amazon Fan sector with water depth deeper than 4000 m. It is interesting that some of the Ad/Al ratios in the study of Feng et al. (2016 JGR) are only 0.04 or 0.1, which correspond to values found in fresh plant tissues. The $\Lambda 8$ values in the marine samples in Feng et al. (2016 JGR) are much lower than our results in the NW and Fan sectors. This means there is increasing loss of lignin as the terrestrial organic material is transported to the deep sea region.

**Changes in manuscript:**
We refer to Feng et al. (2016) in the second and fourth paragraph in section 5.2.1 of the revised manuscript.

**Comments from R. Sparkes:**
Page 9 Line 10:
The phrase "reflects the characteristics of the tributaries (Fig 2A)" was a bit confusing. After reading through the following section it seems that dilution of carbon by mineral sediment is the main reason for changes in TOC. Authors could consider being more explicit about this at the start of the section.

**Response:**
Usually the tributaries of the Amazon River are classified by their water colors. Our intention was to say that the TOC contents reflect the features of tributaries with different colors. The color of a tributary is influenced be the predominance of either dissolved organic matter or suspended

sediment.

**Changes in manuscript:**
Rephrased. The new sentence reads "The distribution of TOC contents basically reflects the characteristics of the tributaries, which are mainly influenced by the content of dissolved organic matter and suspended sediment (Fig. 2A)" (page 9, line 9-11of the revised paper).

**Comments from R. Sparkes:**
Page 9, Line 28:
A good relation is claimed between $\Lambda 8$ and $\Sigma 8$. Authors could consider referring to a figure or providing the p value?

**Response:**
The $\Lambda 8$ and $\Sigma 8$ values for individual tributaries (except for the Xingu River) are correlated. We do not think that an additional figure is necessary here, as the relationship is rather straightforward and also to be expected for most settings. Therefore we only report the average $r^2$=value (average r2=0.76, p<0.05, n=39). The p-values for these correlations are 0.02 for the Negro River, 0.03 for the Solimoes River, $7.0 \times 10^{-4}$ for the Madeira River and $6.8 \times 10^{-6}$ for the Amazon mainstream. All p-values are all smaller than 0.05, which is given with the average r2 value.
Probably we should not show the size of the sample (n=39) here, because it is not the size for individual tributaries. We will delete it in the modified version. We will also show the $r^2$ value of Xingu River ($r^2$=0.02, $p$=0.71) to compare.

**Changes in manuscript:**
R2 value of the Xingu River is provided in the first sentence in section 5.1.2.

**Comments from R. Sparkes:**
Page 12, Line 15: Authors could use a simple mixing model to estimate the proportion of marine and terrestrial carbon in their offshore sediments.

**Response:**
This is indeed a good way to estimate the contribution of marine and terrestrial organic carbon in the offshore sediments. However, the focus of this manuscript is on investigating the distribution of terrigenous OC represented by lignin in the lowland Amazon basin and offshore and the effect of grain size on the distribution. Estimating the proportion of marine and terrestrial contributions in the marine sediments is beyond the scope of this study and would only add another side aspect.

**Changes in manuscript:**
No changes conducted.

**Comments from R. Sparkes:**
Page 12, Line 25:
$\Lambda 8$ and d13C are positively correlated. Authors could show this using p values or a figure.

**Response:**

Figure 6 shows the positive relation between $\Lambda 8$ and d13C of all samples, the $r^2$ and p-value have been added.

**Changes in manuscript:**

r2 and p values for all samples ($r^2$=0.53, $p$<0.05, $n$=27) and for samples from Fan and NW sectors ($r^2$=0.58, $p$<0.05, $n$=21) are added in section 5.2.1.

**Comments from R. Sparkes:**

Page 13, Line 28:
Sample GeoB16225-2 is 30 km offshore, 30 km from a small local river, and in only 34 m water depth with a fine sand sedimentology. Can the authors be confident that this sample reflects material from the Amazon and not local input?

**Response:**

The sample GeoB16225-2 is located about 30 km away from the Sinnamary River mouth and located far away from the Amazon River mouth. Although the Brazil-Guianas coastal mud belt is about 1800 km long and the mudbanks on the inner shelf extend seaward to the limit of the coastal mud wedge (20-25 m water depth) (Allison and Lee, 2004, Mar. Geol. 163), which reaches the location of GeoB16225-2, we can not exclude a contribution from a local river. Nevertheless, given the basically constant lignin content over the lowland Amazon basin, the organic matter from the small local river likely has similar characteristics as the Amazon River system, which makes a distinction from Amazon-derived material virtually impossible.

**Changes in manuscript:**

This problem is pointed out and discussed in the second paragraph of section 5.2.2.

**Typographical changes:**

Page 6 Line 13. "were". The rest of this paragraph is in the present tense
Page 7 Line 15. "varied" The rest of this paragraph is in the present tense
Page 9 Line 19. "‰ )". There is an extra space
Page 9 Line 20. "Bouchez et al. , 2014". There is a space in the wrong location
Page 12 Line 24. "reaching very low $\Lambda 8$ value". It is unclear whether the authors mean to use the singular or plural form "reaching a very low $\Lambda 8$ value"/"reaching very low $\Lambda 8$ values"

**Response:**

Thanks for pointing out these typos.

**Changes in manuscript:**

These mistakes are corrected accordingly.

---

## Author Comment (AC2) · 15 Mar 2017

The comment was uploaded in the form of a supplement:
http://www.biogeosciences-discuss.net/bg-2016-468/bg-2016-468-AC2-supplement.pdf
* * *